# Iterative Refinement Neural Operators Are Learned Fixed-Point Solvers: A Principled Approach to Spectral Bias Mitigation

Xiaotian Liu [1]   Shuyuan Shang [2]   Xiaopeng Wang [1]   Pu Ren [3]   Yaoqing Yang [1]

## Abstract

Neural operators serve as fast, data-driven surrogates for scientific modeling but typically rely on a monolithic, single-pass inference procedure that struggles to resolve high-frequency details, a limitation known as spectral bias. We introduce the Iterative Refinement Neural Operator (IRNO), which augments pre-trained operators with a learned refinement module iteratively applied via fixed-point iteration. IRNO decomposes the prediction into a coarse initialization followed by successive residual corrections, paralleling classical numerical solvers. Under mild assumptions, we establish contraction of the induced operator, ensuring convergence to a unique fixed point. To explicitly target high-frequency errors, we propose a progressive spectral loss that adaptively increases penalty on high-frequency components over refinement steps during training. Across physical systems, IRNO consistently lowers error, with up to 56.05% improvement on turbulent flow. On Active Matter, spectral analysis reveals that, relative to base operator, the normalized error ratios decrease to 27.72–36.10% in low-, 5.07–6.68% in mid-, and 1.48–2.04% in high-frequencies, remaining stable beyond the trained iteration count.

## 1. Introduction

Neural operators have emerged as an effective approach for learning mappings between function spaces, enabling fast surrogate models for parametric Partial Differential Equations (PDEs) and complex physical systems. Architectures such as the Fourier Neural Operator (FNO) (Li et al.,

2021) and DeepONet (Lu et al., 2021) achieve high accuracy across a range of scientific tasks while maintaining tractable inference computation costs. In this work, we introduce the **Iterative Refinement Neural Operator (IRNO)**, a learned refinement operator that augments a pre-trained neural operator with iterative application of a shared-weight update as a test-time inference loop. Instead of producing a single solution estimate, IRNO defines a dynamical process in function space that progressively corrects residual error. IRNO enables consistent error reduction across refinement steps, remains stable when evaluated for iteration counts beyond those used during training, and improves spectral precision without retraining the base model.

Neural operator architectures are typically trained to approximate the solution operator through a single forward evaluation. Improvements in accuracy are therefore primarily achieved through *training-time scaling*, including increased model capacity, higher-resolution data, or larger training sets. Unlike monolithic operators, IRNO decouples accuracy improvement from retraining through *test-time iteration*. Furthermore, the function-space formulation allows IRNO trained with one base operator to refine predictions from different operators.

From the perspective of numerical analysis, this refinement process defines a learned fixed-point iteration in function space. IRNO applies a sequence of residual corrections that progressively reduce the remaining error, aligning the method with classical residual-based solvers (Almgren et al., 2013) while preserving the expressiveness and efficiency of modern neural operators.

We provide a theoretical analysis showing that IRNO converges as a contraction mapping in function space in Section 3. The analysis models each refinement update as a locally affine map in a neighborhood of the solution manifold. We establish conditions for monotonic convergence toward a unique fixed point and show presence of a residual floor when bias is present. These results provide a formal interpretation of the convergence and saturation behavior when extrapolated beyond training iterations.

Across systems and tasks, IRNO yields consistent error reduction across refinement steps, strong improvements in

[1]Dartmouth College, Hanover, NH, USA [2]The Chinese University of Hong Kong, Shenzhen, China [3]Lawrence Berkeley National Lab, Berkeley, CA, USA. Correspondence to: Yaoqing Yang <Yaoqing.Yang@dartmouth.edu>.

*Proceedings of the $43^{rd}$ International Conference on Machine Learning*, Seoul, South Korea. PMLR 306, 2026. Copyright 2026 by the author(s).

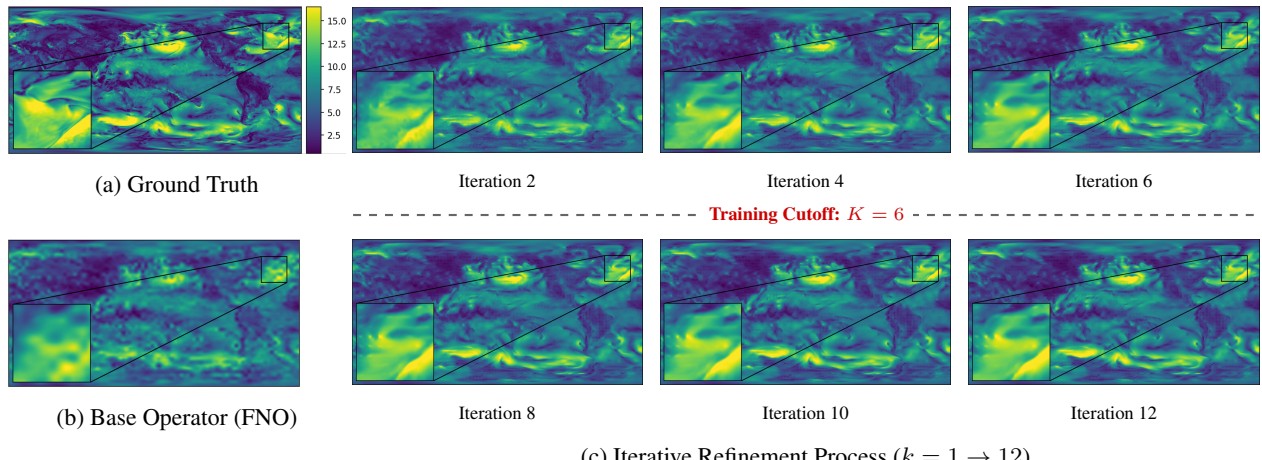

(a) Ground Truth    Iteration 2    Iteration 4    Iteration 6

Training Cutoff: $K = 6$

(b) Base Operator (FNO)    Iteration 8    Iteration 10    Iteration 12

(c) Iterative Refinement Process ($k = 1 \rightarrow 12$)

*Figure 1.* **Iterative Refinement on ERA5 16× Super-Resolution.** The base operator (FNO) captures large-scale atmospheric structure but under-resolves fine-scale details. Successive refinement steps ($k = 1 \rightarrow 12$) from IRNO progressively reduce high-frequency error while preserving global coherence. Horizontal dashed line indicates the training cutoff at $K = 6$; iterations beyond this point demonstrate stable extrapolation and continue to recover fine-scale details. All panels share a common color scale. Shown field: kinetic energy.

mid-to-high frequency errors, stable extrapolation to iteration counts up to twice those used during training, transferability across base operators, and Pareto-dominant accuracy–compute trade-offs relative to capacity-matched monolithic baselines. Figure 1 illustrates the refinement process on ERA5 (Hersbach et al., 2020; Ren et al., 2025), where successive iterations recover fine-scale structure while preserving large-scale coherence.

Our contributions are summarized as follows:

1. **Iterative Refinement Operator.** We introduce a modular, architecture-agnostic iterative refinement operator that enhances predictions of pre-trained neural operators, enabling multi-step error reduction at inference time.

2. **Contraction-Based Analysis.** We model refinement as a fixed-point iteration in function space and derive conditions for monotonic convergence, stable extrapolation over iteration, and a bias-controlled residual error floor, which we empirically validate by correlating the fixed-point bias with the observed error plateau in Figure 3.

3. **Consistent Error Reduction, Spectral Bias Mitigation, and Cross-Operator Transfer.** We conduct a systematic empirical study on efficacy of IRNO across multiple physical systems in Section 4. IRNO achieves consistent error reductions across benchmarks (Section 4.2), demonstrates effective spectral bias mitigation (Section 4.3), exhibits cross-operator transferability between distinct neural operator architectures (Section 4.5), and extends effectively to unstructured mesh settings (Section 4.6). Cost–performance analysis (Sec-

tion 4.8) shows that IRNO traces a Pareto-dominant frontier relative to capacity-matched baselines.

Detailed discussions of related work are available in Appendix B. Our code is available at `https://github.com/xiaotianliu-dartmouth/Iterative_Refinement_Neural_Operator`.

## 2. Methodology

In this section, we formalize IRNO as an inference-time fixed-point scheme in which a pre-trained base operator provides an initial estimate that is iteratively refined by residual correction. Let $\mathcal{X}$ and $\mathcal{H}$ be Banach spaces representing the input (e.g., initial conditions) and solution function spaces, respectively. Our goal is to learn the solution operator $\mathcal{G} : \mathcal{X} \rightarrow \mathcal{H}$ from a finite dataset $\mathcal{D} = \{(x_i, y_i)\}_{i=1}^N$, where $y_i = \mathcal{G}(x_i)$.

### 2.1. Iterative Refinement Neural Operator (IRNO)

2.1.1. INFERENCE-TIME REFINEMENT SCHEME

The inference process is a two-stage dynamical system:

1. **Initialization:** A pre-trained base operator $T_{\text{base}} : \mathcal{X} \rightarrow \mathcal{H}$ (e.g., FNO, DeepONet) produces an initial coarse ansatz $h_0 = T_{\text{base}}(x)$. This captures the dominant low-frequency structure but may lack local fidelity.

2. **Iterative Correction:** A learned refinement operator $\Phi_\theta : \mathcal{X} \times \mathcal{H} \rightarrow \mathcal{H}$ estimates the local residual. The solution state is updated via the fixed-point iteration:

$$h_{k+1} = h_k + \alpha \cdot \Phi_\theta(x, h_k), \quad k = 0, \dots, K - 1,$$

where $\alpha \in (0, 1]$ is a step size of choice, controlling the trade-off between convergence speed and stability.

This formulation enables $\Phi_\theta$ to refine predictions from different base operators $T_{\text{base}}$ (Section 4.5) and allows us to establish IRNO as a contraction mapping in function space (Section 3). The comparison between the workflows of a single-pass and the iterative scheme is shown in Figure 2.

**Architectural Choice.** The refinement operator $\Phi_\theta$ is designed to represent a stable update rule that maps the current iterate $h_k$ to a corrective residual, rather than directly approximating the solution operator itself. Architectural considerations are therefore guided by numerical properties of the induced inference dynamics: (i) *smoothness* to ensure stable iteration, (ii) *multi-scale expressiveness* to capture spectral corrections, and (iii) *computational efficiency* through weight sharing across iterations.

These requirements can be satisfied by various multi-scale encoder-decoder architectures with skip connections. In our experiments (Section 4), we instantiate $\Phi_\theta$ as a lightweight U-Net (Ronneberger et al., 2015), though the framework is architecture-agnostic and compatible with any operator backbone that meets these design criteria.

## 2.2. Training the Refinement Operator

We train $\Phi_\theta$ using a composite objective that enforces trajectory control, frequency recovery, and convergence stability.

### 2.2.1. MULTI-STEP SUPERVISION (TRAJECTORY CONTROL)

To ensure the iterative process creates a stable dynamical trajectory toward the solution, we impose supervision on the output of every refinement step. We minimize the $L^2$ error norm averaged over the trajectory:

$$\mathcal{L}_{\text{spatial}} = \frac{1}{K} \sum_{k=1}^{K} \|h_k - y\|^2.$$

This deep supervision prevents the network from learning unstable intermediate updates that might drift off the solution manifold before being corrected. Appendix E demonstrates that without trajectory control, the learned operator fails to satisfy the strong monotonicity condition underlying the contraction theory (Section 3).

### 2.2.2. PROGRESSIVE SPECTRAL LOSS (FREQUENCY RECOVERY)

Standard $L^2$ loss often fails to capture high-frequency details. We apply a progressive spectral loss that dynamically pivots from coarse to fine scales as iterations proceed. Early refinement steps focus on correcting coarse structure, while

### a) Single Forward Pass Neural Operators

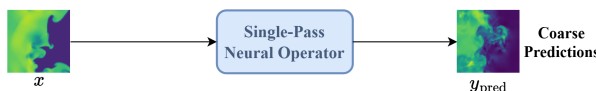

### b) Iterative Refinement Neural Operators (IRNO)

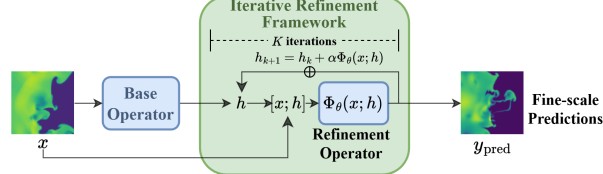

*Figure 2.* **Overview of the Iterative Refinement Neural Operator (IRNO).** (a) Standard neural operators approximate the solution in a monolithic, single forward evaluation, often losing fine-scale details. (b) IRNO reformulates inference as a dynamic process. A Base Operator provides a coarse initialization, which is then iteratively corrected by a shared-weight refinement operator $\Phi_\theta$. At each step $k$, the network concatenates the original input $x$ with the current estimate $h_k$ to predict a residual update, progressively resolving fine-scale local physics.

later steps progressively recover local features, implementing multi-resolution numerical refinement.

Let $\hat{h}_k = \mathcal{F}(h_k)$ be the Fourier transform of the estimate. The spectral loss at step $k$ is defined as

$$\mathcal{L}_{\text{spectral}}^{(k)} = \frac{1}{HW} \frac{1}{\bar{\rho}_k} \sum_\omega \rho(\omega, \lambda_k) \cdot \left| |\hat{h}_k(\omega)| - |\hat{y}(\omega)| \right|^2,$$

where and the weighting function $\rho$ penalizes high frequencies exponentially as $\rho(\omega, \lambda_k) = 1 + (|\omega|/|\omega|_{\text{nyq}})^{\lambda_k}$, and $\bar{\rho}_k$ is the averaged weights. The exponent $\lambda_k$ increases linearly from $\lambda_{\text{start}}$ to $\lambda_{\text{end}}$ across iterations. The total spectral loss is the average over the trajectory,

$$\mathcal{L}_{\text{spectral}} = \frac{1}{K} \sum_k \mathcal{L}_{\text{spectral}}^{(k)}.$$

### 2.2.3. FIXED-POINT REGULARIZATION (CONVERGENCE STABILITY)

Finally, we explicitly enforce the stability condition of a fixed-point solver. If the input is the exact solution $y$, the residual correction should be zero. We add a regularization term

$$\mathcal{L}_{\text{fp}} = \|\Phi_\theta(x, y)\|^2.$$

This ensures that $y$ is a fixed point of the learned dynamics, preventing the model from pushing the state away from the true solution once converged. The fixed-point regularization $\mathcal{L}_{\text{fp}}$ directly minimizes the bias term in Theorem 3.1, thereby reducing the error floor from Corollary 3.3.

**Total Objective.** The full loss function is a weighted sum:

$$\mathcal{L}_{\text{total}} = \mathcal{L}_{\text{spatial}} + \beta_{\text{spectral}} \mathcal{L}_{\text{spectral}} + \beta_{\text{fp}} \mathcal{L}_{\text{fp}},$$

where $\beta_{\text{spectral}}$ and $\beta_{\text{fp}}$ are hyperparameters balancing spectral corrections, convergence stability, and spatial fidelity.

Importantly, $\Phi_\theta$ learns the iteration-invariant *update dynamics* rather than the solution mapping itself and can be applied for $k > K$ steps at inference time.

# 3. Theoretical Analysis

We analyze the convergence behavior of the proposed iterative refinement operator $\Phi(x, \cdot)$. Let $e_k = y - h_k$ denote the residual error at iteration $k$. The analysis characterizes how $\|e_k\|$ evolves under mild regularity assumptions on the learned refinement operator $\Phi(x, \cdot)$ and the pre-trained base operator $T_{\text{base}}$.

## 3.1. Theoretical Setup

Formally, we assume the following properties hold in a local neighborhood of the true solution $y$:

1. **Local Affine Approximation:** In a neighborhood $B_\delta(y) = \{h_i \in \mathcal{H} : \|e_i\| < \delta\}$ of the true solution, we assume the refinement operator $\Phi(x, \cdot)$ admits an affine linearization of the form

   $$\Phi(x, h) = b(x) + A(x, h)e + R(x, h),$$

   where $e = y - h$, $A(x, h) : \mathcal{H} \to \mathcal{H}$ is a bounded linear operator, $b(x)$ is the bias term, and the remainder $R(x, h)$ satisfies $\|R(x, h)\| \leq \frac{L}{2}\|e\|^2$, for some $L > 0$.

   For notational simplicity, we suppress the $x$ dependence and write $b(x)$ as $b$. By evaluating the affine approximation $\Phi(x, h)$ at $h = y$, we have the bias term $b = \Phi(x, y)$. Ideally, a perfectly learned refinement operator would satisfy $\|b\| = 0$.

   This decomposition is analogous to a first-order Taylor expansion of $\Phi(x, \cdot)$ around the true solution $y$, with $A(x, h)$ being a linear operator acting on residual $e$ and $R(x, h)$ capturing higher-order remainder terms.

2. **Lipschitz continuity of the operator-valued map and strong monotonicity:** The mapping $h \mapsto A(x, h)$ should vary smoothly in a neighborhood of the fixed point. Assume there exists $\mu > 0$ such that for all $h \in B_\delta(y)$,

   $$\|A(x, h) - A(x, y)\|_{\text{op}} \leq \mu\|e\|$$

   where $A(x, y) \equiv A(x)$.

   Furthermore, we assume the linearization at the solution, $A(x, y)$, is locally bounded and strongly monotone, i.e., there exist constants $0 < m \leq M < \infty$ such that

   $$\langle A(x, y)e, e \rangle \geq m\|e\|^2, \quad \|A(x, y)\|_{\text{op}} \leq M.$$

Under this condition, choosing $0 < \alpha < \frac{2m}{M^2}$ guarantees $\|I - \alpha A(x, y)\|_{\text{op}} = q < 1$, since for any unit vector $e$,

$$\|(I - \alpha A)e\|^2 \leq 1 - 2\alpha m + \alpha^2 M^2 < 1.$$

The Lipschitz continuity assumption on $h \mapsto A(x, h)$ is standard in convergence analysis (Werner & Hofmann, 2019; Rastogi et al., 2020; Kovachki et al., 2023), while the strong monotonicity condition is empirically validated in Appendix E and ensures the spectral radius condition $q < 1$ is achievable by explicit choices of $\alpha$.

3. **Initialization quality and invariant-ball:** The base operator $T_{\text{base}}$ provides a sufficiently accurate initial ansatz such that

   $$\|e_0\| < \min\left\{\delta, \frac{1-q}{2c}\right\}, \quad c = \alpha\left(\frac{L}{2} + \mu\right),$$

   where $\delta$ is the radius in Assumption 1. We further assume there exists $r$ such that $\|e_0\| \leq r < \min\left\{\delta, \frac{1-q}{2c}\right\}$ and

   $$\alpha\|b\| \leq r(1 - q - cr),$$

   which is a small-bias condition ensuring the iterates remain in $B_r(y) \subset B_\delta(y)$. This bias magnitude is directly minimized by the fixed-point regularization $\mathcal{L}_{\text{fp}}$ in Section 2.2.3. A sufficient condition and derivation are given in Appendix F.

Intuitively, Assumption 1 reflects the fact that $\Phi$ is trained by deep supervision to approximate the residual $e = y - h$, and its local linearization is encouraged to produce residual-correcting updates near the solution manifold. Assumption 2 captures architectural smoothness induced by convolutional layers in $\Phi$, which empirically limit abrupt changes in the refinement dynamics. The strong monotonicity condition further ensures that a valid step size $\alpha$ exists, and is empirically validated in Appendix E. Finally, Assumption 3 formalizes the role of $T_{\text{base}}$ as providing a reasonable initialization within the basin of attraction, and requires the learned bias $\|b\|$ to remain small enough to keep iterates from escaping the local neighborhood, a condition directly enforced by the fixed-point regularization $\mathcal{L}_{\text{fp}}$ in Section 2.2.3.

## 3.2. Main Results

**Theorem 3.1** (Quadratic–Linear Convergence). *Under the above assumptions, the iteration satisfies*

$$\|e_{k+1}\| \leq q\|e_k\| + c\|e_k\|^2 + \alpha\|b\|.$$

*When $b = 0$, the scheme is locally contractive:*

$$\|e_{k+1}\| \leq (q + c\|e_k\|)\|e_k\|, \quad \text{with } q + c\|e_k\| < 1.$$

*Thus, the errors decrease monotonically and $h_k \to y$.*

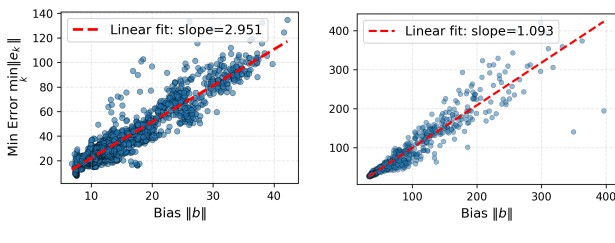

(a) Active Matter (FNO)          (b) TR-2D (TFNO)

*Figure 3.* **Empirical validation of bias–error floor relationship.** Scatter plots of minimum error $\min_k \|e_k\|$ as a function of bias magnitude $\|b\| = \|\Phi_\theta(x, y)\|$ over $k = 24$ refinement steps. **(a)** Active Matter: Pearson $r = 0.933$ ($p \ll 10^{-10}$). **(b)** TR-2D: Pearson $r = 0.949$ ($p \ll 10^{-10}$). Both systems exhibit a strong linear dependence between the asymptotic error floor and the bias, consistent with Corollary 3.3.

*Table 1.* **Iterative error reduction across physical systems.** Comparison of single-pass base operator performance and IRNO after refinement, with FNO evaluated at $K = 6$ and TFNO/WDSR at $K = 4$. Metrics reported are VRMSE for TR-2D and Active Matter, and ACC and RFNE for ERA5 with standard errors $< 0.001$ for all entries. ↑/↓ indicate better direction.

| Dataset | Metric | Base Model | Initial | IRNO (Ours) | Improvement(%) |
|---|---|---|---|---|---|
| TR-2D | VRMSE ↓ | FNO | 0.2394 | **0.1309** | 45.32% |
| | | TFNO | 0.2371 | **0.1042** | 56.05% |
| Active Matter | VRMSE ↓ | FNO | 0.1017 | **0.0501** | 50.73% |
| | | TFNO | 0.1981 | **0.0387** | 80.46% |
| ERA5 | ACC ↑ | FNO | 0.7523 | **0.8919** | 18.56% |
| | | WDSR | 0.9091 | **0.9104** | 0.143% |
| | RFNE ↓ | FNO | 0.3247 | **0.2140** | 34.09% |
| | | WDSR | 0.2119 | **0.1953** | 7.83% |

Proof of Theorem 3.1 is available in Appendix C. The result in Theorem 3.1 characterizes IRNO as a learned iterative solver whose convergence behavior is governed by the contraction factor $q$, smoothness of its update dynamics, and the bias, providing a theoretical basis for the monotonic error decay and stable extrapolation observed empirically in Section 4. Detailed interpretations of the error recursion in Theorem 3.1 are provided in Appendix C.

**Corollary 3.2** (Geometric Convergence and Iteration Complexity). *In the linear-dominant regime with $b = 0$, the error decays geometrically:*

$$\|e_k\| \lesssim q^k \|e_0\|.$$

*To achieve $\|e_k\| \leq \varepsilon$, it suffices to perform*

$$k = O\left(\frac{\log(\|e_0\|/\varepsilon)}{\log(1/q)}\right)$$

*iterations of refinement.*

Corollary 3.2 assumes $b = 0$, allowing arbitrarily small error with sufficient iterations. In practice, however, $\Phi_\theta$ may exhibit non-zero bias $b \neq 0$, introducing a limiting error floor. The following corollary quantifies the upper bound of the error floor.

**Corollary 3.3** (Convergence with Bias). *If $b = \Phi(x, y) \neq 0$ and $\|b\|$ satisfies the invariant-ball condition, then there exists a unique fixed point $h^*$ satisfying $h^* = h^* + \alpha\Phi(x, h^*)$, and the iteration converges linearly to $h^*$ for any initialization within a neighborhood of $y$. The limiting error satisfies*

$$\|e^*\| \leq \frac{\alpha\|b\|}{1 - q} + O(\|b\|^2).$$

Proofs of Corollaries 3.2 and 3.3 are available in Appendix D. Under mild assumptions, the refinement map $T(h) = h + \alpha\Phi(x, h)$ is locally contractive. The contraction

factor $q$ governs the convergence speed, directly linking empirical accuracy to the choice of step size $\alpha$ (Section 4.7.3). In presence of non-zero bias, convergence remains guaranteed with a limiting error floor proportional to $\|b\|$.

Empirical validation in Figure 3 supports the conclusions of Corollary 3.3 across two physical systems (Section 4.1). For both Active Matter and TR-2D, the minimum attainable error over the refinement trajectory exhibits a strong linear dependence on the bias magnitude, with Pearson correlations exceeding 0.93 in both cases ($p \ll 10^{-10}$). Least-squares fits indicate that variation in the asymptotic error floor is largely explained by the magnitude of the fixed-point residual $b$, motivating fixed-point regularization to directly reduce $\|b\|$ (Section 2.2.3).

## 4. Experiments

We evaluate IRNO from the perspective of a learned iterative solver rather than a single-pass predictor. Experiments are designed to assess six core areas: (i) global convergence behavior across distinct physical systems and extrapolation beyond the training iteration cutoff, (ii) spectral dynamics of refinement and preferential reduction of mid-to-high frequency spectral error, (iii) transferability across base operators and operator architectures, (iv) the role of progressive spectral supervision and step size $\alpha$ through targeted ablations, (v) the cost–performance trade-off relative to capacity scaling, and (vi) generalization to graph-based operators on irregular meshes.

### 4.1. Experimental Setup

**Physical Systems.** We evaluate IRNO on four scientific benchmarks: (i) Turbulent Radiative Layer (TR-2D) and (ii) Active Matter from the Well (Ohana et al., 2024; Morel et al., 2025; Wu et al., 2025; Holzschuh et al., 2025), (iii) ERA5 global weather $16\times$ super-resolution from SuperBench (Ren et al., 2025; Chen et al., 2024; Hassan et al., 2023), and (iv) CE-Gauss (Mousavi et al., 2025), an irregular unstructured-

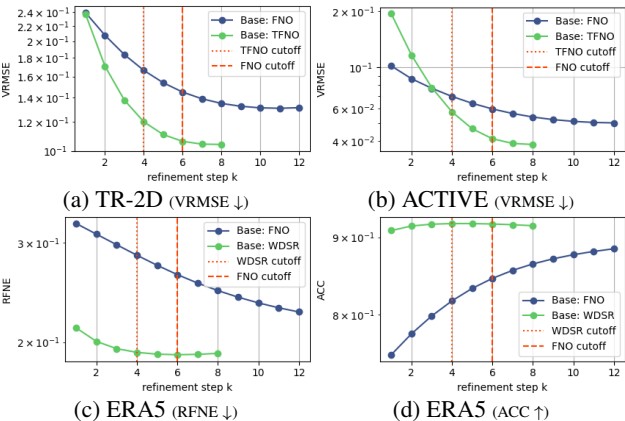

(a) TR-2D (VRMSE ↓)  (b) ACTIVE (VRMSE ↓)

(c) ERA5 (RFNE ↓)  (d) ERA5 (ACC ↑)

*Figure 4.* **Convergence behavior across physical systems.** VRMSE for TR-2D and Active Matter, and RFNE and ACC for ERA5, plotted as a function of refinement step $k$. Vertical dashed lines indicate the training cutoff (FNO at $K = 6$ and TFNO/WDSR at $K = 4$). Error metrics decrease or reach a stable plateau for $k > K$, while ACC increases and stabilizes.

mesh benchmark. These datasets are widely adopted for assessing operator stability, multi-scale error dynamics, and high-resolution reconstruction. Detailed dataset description can be found in Appendix G.1.

We report variance-scaled RMSE (VRMSE) for TR-2D and Active Matter, anomaly correlation coefficient (ACC) and relative Frobenius norm error (RFNE) for ERA5, and relative $L^2$ error for CE-Gauss. Definitions of the metrics are detailed in Appendix G.5.1.

**Base Operators and Baselines.** For TR-2D and Active Matter, we use the FNO (Li et al., 2021) and the Tucker-Factorized FNO (TFNO) (Kossaifi et al., 2025) as base operators. For ERA5, we evaluate both FNO and the Wide-Activated Deep Super-Resolution network (WDSR) (Fan et al., 2018). For CE-Gauss, we implement RIGNO (Mousavi et al., 2025).

We compare IRNO against three baselines. **(i) Standard Base Operator:** single-pass inference using the pre-trained backbone without refinement. **(ii) Capacity-Matched Residual Model:** single-shot residual correction networks with parameter count and FLOPs exceed those of IRNO executed for $K$ refinement steps. This baseline controls for improvements due to increased model capacity rather than iterative refinement. **(iii) State-of-the-Art Models for Spectral Bias Mitigation:** Hierarchical Neural Operator Transformer (HiNOTE) (Luo et al., 2024) and High Frequency Scaling (HFS) (Khodakarami et al., 2026), compared on ERA5 in Table 2.

**Training and Inference.** The refinement operator is trained with a horizon of $K = 4, 6$ steps using composite objective defined in Section 2.2. The base operator $T_{\text{base}}$ is pre-trained

*Table 2.* **ERA5 $16\times$ super-resolution against spectral baselines.** Comparison of IRNO against state-of-the-art spectral methods on ERA5, with IRNO evaluated using WDSR as the base operator. Metrics reported are ACC and RFNE. ↑/↓ indicate better direction.

| Method | ACC ↑ | RFNE ↓ |
|---|---|---|
| ResUNet-HFS (Khodakarami et al., 2026) | 0.8915 | 0.2253 |
| HiNOTE (Luo et al., 2024) | 0.9055 | 0.2222 |
| IRNO (WDSR, Ours) | **0.9104** | **0.1953** |

and frozen. Models trained with $K = 4$ are evaluated for $k \in [0, 8]$, and models trained with $K = 6$ are evaluated for $k \in [0, 12]$. Detailed experiments setup is available in Appendix G.

### 4.2. Global Convergence Behavior Across Physical Regimes

We evaluate whether IRNO performs stable and monotonic error reduction across distinct physical systems, consistent with the contractive dynamics predicted by Theorem 3.1.

**Quantitative Performance.** Table 1 reports aggregate performance metrics for TR-2D, Active Matter, and ERA5. Across all benchmarks and base architectures, IRNO consistently improves upon base operator inference, achieving 45–56% reductions on TR-2D, 51–80% on Active Matter in VRMSE and 27% in RFNE while increasing 18% ACC on ERA5. Gains on WDSR are less significant (ACC 0.14%↑, RFNE 7.83%↓), reflecting its already strong super-resolution performance (ACC≈0.91, RFNE≈0.21 (Ren et al., 2025)), leaving less residual structure for correction.

**Error Trajectories vs. Iteration.** Figure 4 shows that error decreases monotonically during early iterations and reaches a stable plateau. Notably, no divergence occurs for $k > K$, indicating stable extrapolation beyond the training cutoff.

**Comparison with State-of-the-Art Models for Spectral Bias Mitigation.** Table 2 compares IRNO against recent spectral methods on ERA5 $16\times$ super-resolution. IRNO (WDSR) achieves the best ACC (0.910) and RFNE (0.195), outperforming both High-Frequency Scaling (HFS) (Khodakarami et al., 2026) and the hierarchical attention operator HiNOTE (Luo et al., 2024). Furthermore, IRNO is complementary to these architectural spectral methods. On Active Matter, combining IRNO with HFS further reduces VRMSE from 0.0631 to 0.0486, suggesting the iterative refinement mechanism compounds gains from frequency-aware architectures. Table 17 and Figure 19 in the appendix provide a detailed step-by-step breakdown, showing VRMSE decreasing from 0.0631 (HFS base) to 0.0486 at $k = 6$ and remaining stable at $k = 8$ (0.0487), with IRNO achieving lower spectral error energy across the full radial frequency range.

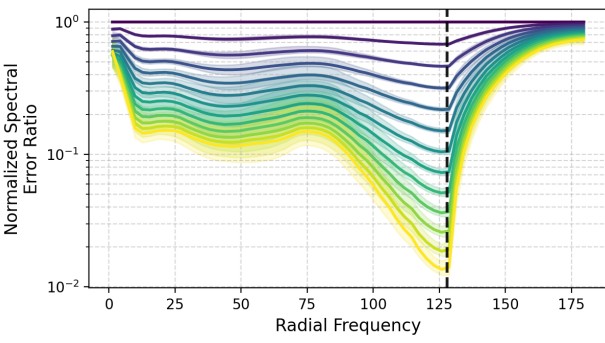

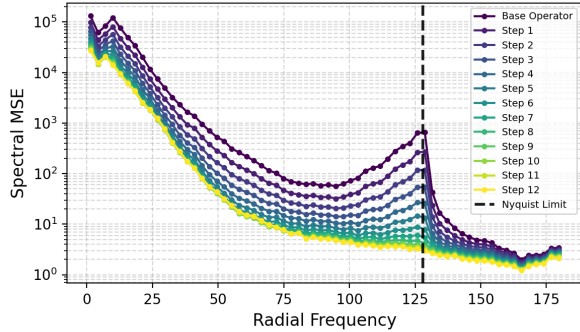

*(a)* Dataset-level median normalized spectral error ratios $\tilde{E}^{(k)}(\omega)$.

*(b)* Instance-level spectral MSE trajectories.

*Figure 5.* **Spectral Error Evolution under iterative refinement (Active Matter, FNO).** *(a)* Median normalized spectral error ratios $\tilde{E}^{(k)}(\omega)$ across the test set, with shaded interquartile ranges (25–75%). IRNO exhibits consistent attenuation of mid-to-high frequency error with increasing refinement steps, with stable behavior near the Nyquist limit $\omega = 128$, indicated by a vertical dashed line. *(b)* Spectral Mean Squared Error (MSE) for a representative test instance for $k \in [0, 12]$ refinement steps. Each curve shows the radial-spectral error of the base operator and successive IRNO refinements, illustrating monotonic reduction of spectral error and stable behavior beyond the training cutoff. Across both panels, mid-to-high frequency error decreases with refinement, with the largest relative reductions observed near the Nyquist limit $\omega = 128$ (vertical dashed line).

## 4.3. Spectral Dynamics of Iterative Refinement

To characterize the spectral dynamics of IRNO, we analyze how error evolves in the frequency domain across refinement steps at both the dataset and instance levels.

### 4.3.1. DATASET- AND INSTANCE-LEVEL SPECTRAL ERROR DISTRIBUTION

For each refinement step $k$ and radial frequency $\omega$, we compute the normalized spectral error ratio

$$\tilde{E}^{(k)}(\omega) = \frac{E^{(k)}(\omega)}{E^{(0)}(\omega) + \varepsilon},$$

where $E^{(0)}(\omega)$ denotes the spectral error of the base operator and $\varepsilon = 10^{-10}$ is a small constant for numerical stability. We report the median and interquartile range (25–75%) across the test set.

Figure 5(a) shows that refinement reduces error primarily in mid-to-high frequency regimes where base operators exhibit strongest spectral bias. We further quantify this observation in Section 4.4. Figure 5(b) confirms monotonic spectral error reduction per-sample with largest improvements near $\omega = 128$ (Nyquist limit).

## 4.4. Frequency-Band Convergence

To summarize frequency-dependent convergence, we partition the spectrum into low-, mid-, and high-frequency bands using resolution-invariant percentile cutoffs (bottom 33%, middle 33%, and top 33%, respectively). For each band $B$ and refinement step $k$, we compute the normalized error ratio aggregated over frequencies in the band and across the test set.

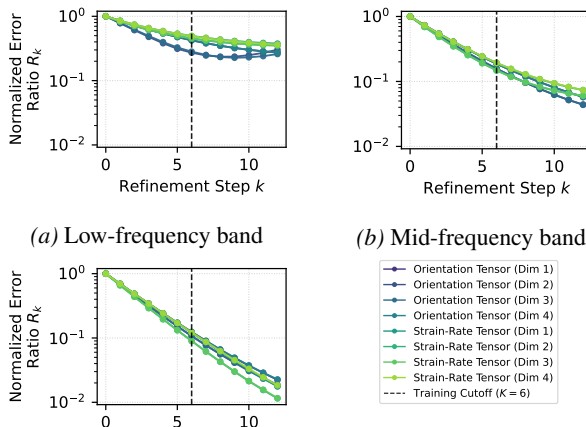

*(a)* Low-frequency band

*(b)* Mid-frequency band

*(c)* High-frequency band

*Figure 6.* **Frequency-band error ratios across refinement steps (Active Matter, FNO).** Normalized error ratios $R_k^{(B)}$ for each tensor field, computed separately over (a) low-, (b) mid-, and (c) high-frequency bands as a function of refinement step $k$. The vertical dashed line marks the training cutoff at $K = 6$. Lower values indicate stronger error reduction relative to the base operator.

Let $\Omega_B$ denote the set of frequencies in band $B$, $i \in \mathcal{D}_{test}$ index test samples, and $E_i^{(k)}(\omega)$ denote the spectral error of sample $i$ at frequency $\omega$ and refinement step $k$. We define

$$R_k^{(B)} = \text{median}_{i \in \mathcal{D}_{test}} \left( \frac{\sum_{\omega \in \Omega_B} E_i^{(k)}(\omega)}{\sum_{\omega \in \Omega_B} E_i^{(0)}(\omega) + \varepsilon} \right).$$

Figure 6 reports $R_k^{(B)}$ across refinement steps for orientation and strain-rate tensor fields. Table 3 summarizes averaged

*Table 3.* **Frequency-band error ratios across refinement steps (Active Matter, FNO).** Normalized error ratios $R_k^{(B)}$ for low-, mid-, and high-frequency bands, reported for orientation and strain tensor fields at the initial step ($k = 1$), training cutoff ($k = 6$), and final step ($k = 12$). Lower values indicate stronger error reduction.

| | | Refinement Step ($k$) | | |
|---|---|---|---|---|
| **Frequency** | **Tensor** | **Initial** ($k = 1$) | **Cutoff** ($k = 6$) | **Final** ($k = 12$) |
| **Low** | Orient. | 0.8145 | 0.3503 | 0.2772 |
| | Strain | 0.8423 | 0.4738 | 0.3610 |
| **Mid** | Orient. | 0.7204 | 0.1722 | 0.0507 |
| | Strain | 0.7120 | 0.1703 | 0.0668 |
| **High** | Orient. | 0.6856 | 0.1157 | 0.0204 |
| | Strain | 0.6789 | 0.1054 | 0.0148 |

*Table 4.* **Transferability across base operators.** IRNO trained on one base operator successfully transfers to refine predictions from different base operators. Subscripts indicate the operator IRNO was trained with (e.g., IRNO$_{\text{TFNO}}$ was trained with TFNO as the base). Standard errors are $< 0.001$ for all entries.

| Dataset | Metric | Base + IRNO$_{\text{train}}$ | Initial | IRNO | Improvement(%) |
|---|---|---|---|---|---|
| TR-2D | VRMSE $\downarrow$ | FNO + IRNO$_{\text{TFNO}}$ | 0.2396 | **0.0994** | 58.53% |
| | | TFNO + IRNO$_{\text{FNO}}$ | 0.2366 | **0.1345** | 43.15% |
| Active Matter | VRMSE $\downarrow$ | FNO + IRNO$_{\text{TFNO}}$ | 0.1004 | **0.0445** | 55.66% |
| | | TFNO + IRNO$_{\text{FNO}}$ | 0.1955 | **0.1127** | 42.36% |
| ERA5 | ACC $\uparrow$ | FNO + IRNO$_{\text{WDSR}}$ | 0.7523 | **0.8022** | 6.22% |
| | | WDSR + IRNO$_{\text{FNO}}$ | 0.9091 | **0.9219** | 1.39% |
| | RFNE $\downarrow$ | FNO + IRNO$_{\text{WDSR}}$ | 0.3247 | **0.2823** | 13.06% |
| | | WDSR + IRNO$_{\text{FNO}}$ | 0.2119 | **0.1935** | 8.68% |

values at the initial, cutoff, and final iterations. Across tensor fields, IRNO achieves 98–99% error reduction in high-frequency bands, 93–96% in mid-frequency bands, and 62–73% in low-frequency bands at the final refinement step.

### 4.5. Transferability Across Base Operators

The function-space formulation of IRNO in Section 2.1 suggests that refinement dynamics depend primarily on the local residual geometry rather than the specific architecture producing the initialization. Under Theorem 3.1, convergence is guaranteed whenever the initialization lies within the basin of attraction (Assumption 3), and the operator is well-conditioned.

As shown in Table 4, IRNO trained on one base operator successfully transfers without retraining to refine predictions from different operators, achieving consistent error reduction across operators and tasks. Notably, IRNO trained on low-performing base operators often outperforms same-operator configurations when transferred to high-performing base operators. For example, IRNO$_{\text{TFNO}}$ improves FNO's initial prediction VRMSE by 58.53% on TR-2D, 13.21 percentage points higher than IRNO$_{\text{FNO}}$'s improvement. This suggests that less accurate operators generate larger, more diverse residual structures during training, forcing IRNO to

learn more robust error-correction strategies that generalize effectively to the smaller, more structured residuals of higher-performing operators.

### 4.6. Generalization to Graph-Based Operators and Irregular Meshes

To evaluate whether IRNO extends beyond structured-grid operators, we apply it to RIGNO (Mousavi et al., 2025), a graph-based neural operator that operates on irregular unstructured meshes. We use the CE-Gauss benchmark, an irregular unstructured mesh dataset with 16,384 nodes and 4 physical variables, and evaluate autoregressive rollout over 7 timesteps with $K = 4$ refinement steps and $\alpha = 0.3$. Dataset details are provided in Appendix G.1.

As shown in Table 5, IRNO reduces $L^2$ error at every timestep, with improvements compounding from 12.5% at $t = 1$ to 21.3% at $t = 7$. This suggests that early refinement suppresses error accumulation in autoregressive rollout. Implementation details are provided in Appendix G.

*Table 5.* **Autoregressive rollout on irregular meshes.** IRNO ($K = 4$) with RIGNO as the base operator on the CE-Gauss unstructured mesh benchmark, evaluated over a 7-step autoregressive rollout. Metric reported is relative $L^2$ error (%) against ground truth. Lower is better.

| Model | $t = 1$ | $t = 2$ | $t = 3$ | $t = 4$ | $t = 5$ | $t = 6$ | $t = 7$ |
|---|---|---|---|---|---|---|---|
| Base (RIGNO) | 3.351 | 5.017 | 7.808 | 10.923 | 12.732 | 14.433 | 16.617 |
| IRNO | 2.931 | 4.359 | 6.605 | 8.990 | 10.371 | 11.450 | 13.080 |
| Improvement | 12.5% | 13.1% | 15.4% | 17.7% | 18.5% | 20.7% | 21.3% |

### 4.7. Ablation Study

#### 4.7.1. PROGRESSIVE VS. FIXED SPECTRAL SUPERVISION

We ablate the proposed progressive spectral loss to evaluate the role of frequency curriculum in stable multiscale refinement. Table 6 compares models trained with a linearly increasing spectral exponent $\lambda_k : 1.0 \rightarrow 2.0$ against fixed-weight baselines with constant $\lambda \in \{1.0, 1.25, 1.75, 2.0\}$.

The progressive spectral loss schedule achieves VRMSE of 0.039 versus 0.051-0.070 for fixed $\lambda$ (23.5-44.3% reduction), with particularly strong high-frequency improvements (normalized error 0.24 vs 0.60-0.88).

#### 4.7.2. ROBUSTNESS ACROSS REFINEMENT ARCHITECTURES AND NORMALIZATION

IRNO's iterative mechanism is robust across refinement backbone choices, with ResNet, ConvNext, and FNO backbones all achieving $> 71\%$ VRMSE reduction on Active Matter (TFNO base, $K = 4$); see Appendix H (Table 18). Normalization choice does not qualitatively affect IRNO's behavior. BatchNorm, LayerNorm, and GroupNorm all

*Table 6.* **Ablation of progressive spectral loss (Active Matter, TFNO).** Comparison of a progressive schedule ($\lambda_k : 1 \to 2$) against fixed spectral weights, with TFNO (4 steps) as base operator. We report VRMSE and frequency-band normalized error ratios for low-, mid-, and high-frequency bands. Lower values indicate stronger error reduction.

| Method | VRMSE | Low-Freq | Mid-Freq | High-Freq |
|---|---|---|---|---|
| Fixed $\lambda = 1.00$ | 0.0509 | 0.0953 | 0.1067 | 0.6023 |
| Fixed $\lambda = 1.25$ | 0.0695 | 0.1599 | 0.2101 | 0.8794 |
| Fixed $\lambda = 1.75$ | 0.0586 | 0.1124 | 0.1320 | 0.6949 |
| Fixed $\lambda = 2.00$ | 0.0666 | 0.2063 | 0.1578 | 0.7677 |
| **Progressive (Ours)** | **0.0387** | **0.0551** | **0.0788** | **0.2393** |

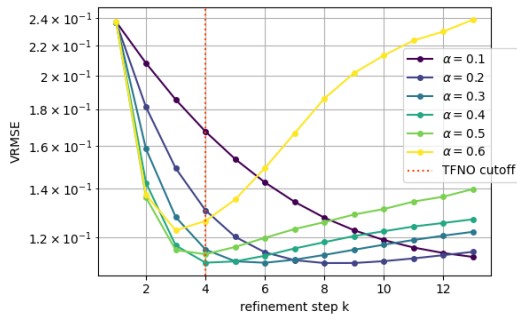

*Figure 7.* **Ablation: Step size $\alpha$ sensitivity (TR-2D, TFNO).** Convergence behavior for $\alpha \in \{0.1, 0.2, 0.3, 0.4, 0.5, 0.6\}$, trained with $K = 4$ steps (dashed line). Small step sizes converge slowly; moderate step sizes ($\alpha \in [0.2, 0.4]$) achieve optimal balance; $\alpha = 0.6$ diverges rapidly beyond the training horizon, violating the contraction condition $q = \|I - \alpha A(x)\|_{\mathrm{op}} < 1$ from Theorem 3.1.

yield consistent error reduction on TR-2D (TFNO base); see Appendix H.

### 4.7.3. VARYING STEP SIZE $\alpha$

Figure 7 shows how step size $\alpha$ affects convergence on TR-2D with TFNO-based IRNO ($k = 4$). Small $\alpha = 0.1$ converges slowly but stably; moderate $\alpha = 0.2$ achieves the best balance of speed and stable extrapolation beyond the training horizon; $0.3 \leq \alpha \leq 0.5$ initially decreases error but diverges for $k > 6$; and $\alpha = 0.6$ diverges within the training horizon (VRMSE $0.12 \to 0.24$). Instability arises when $\alpha$ exceeds $1/\|A(x)\|_{\mathrm{op}}$, violating the contraction condition $q = \|I - \alpha A(x)\|_{\mathrm{op}} < 1$ from Theorem 3.1. We use $\alpha \in \{0.2, 0.25\}$ in our experiments.

### 4.8. Cost–Performance Pareto Frontier

Finally, we evaluate whether iterative refinement is more cost-efficient than scaling model capacity. Figure 8 compares IRNO against capacity-matched monolithic residual models on ACC and RFNE versus FLOPs and memory. IRNO traces a Pareto-dominant frontier across all metrics: at 1100 GFLOPs, IRNO attains ACC = 0.84 versus 0.79 for

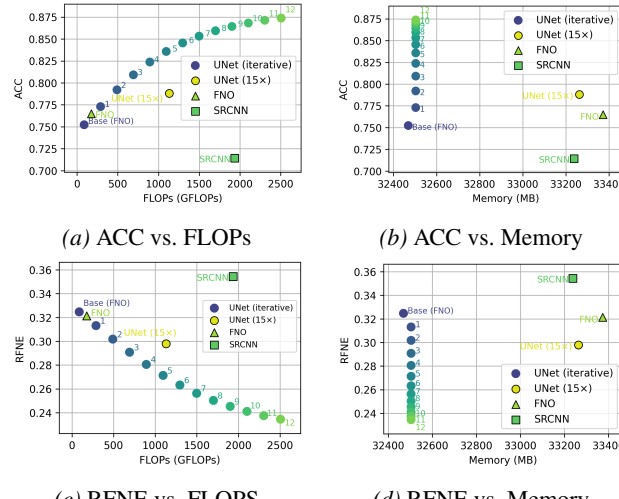

*(a)* ACC vs. FLOPs      *(b)* ACC vs. Memory

*(c)* RFNE vs. FLOPS      *(d)* RFNE vs. Memory

*Figure 8.* **Cost–performance trade-offs under iterative refinement (ERA5, FNO).** ACC and RFNE plotted against computational cost (FLOPs) and memory usage during inference. Each marker corresponds to a refinement step of the iterative U-Net, with step indices annotated. Baselines (FNO, SRCNN, and 15× U-Net) are shown for reference. IRNO achieves improved ACC or lower error at comparable or lower computational cost. See Appendix H for detailed comparisons and wall-clock times.

a 15× U-Net at comparable cost; at 1000 GFLOPs, IRNO reduces RFNE to 0.28 while monolithic baselines remain above 0.295. Gains are therefore driven by the refinement mechanism rather than increased capacity.

We additionally compare against F-Adapter (Zhang et al., 2026), a parameter-efficient spectral fine-tuning method. F-Adapter achieves a 2.31% VRMSE reduction at low computational overhead, with gains concentrated in mid- and high-frequency bands; IRNO trades additional training cost for substantially larger gains (50.73%), representing complementary design regimes. A full spectral comparison is provided in Appendix H.

## 5. Conclusion

We propose IRNO, a learned iterative refinement operator that progressively reduces prediction error across physical systems. We provide a contraction-based theoretical analysis that interprets refinement as a fixed-point iteration in function space, establishing conditions under which monotonic convergence and stability beyond the training horizon are expected. Experimental results validate that IRNO achieves consistent spectral error reduction, stable extrapolation to iteration counts beyond those seen in training, and improved cost–performance trade-offs relative to capacity-matched baselines.

## Acknowledgment

We thank our colleagues and funding agencies. This work is supported by the DARPA AIQ program, the U.S. Department of Energy under Award Number DE-SC0025584, the Allocation Year 2026 DOE Mission Science award, and Dartmouth College.

## Impact Statement

This work advances the accuracy and stability of neural operators for scientific and engineering applications, supporting better modeling of complex systems such as climate dynamics, fluid flows, and multi-physics processes. More accurate surrogate models can accelerate scientific discovery, reduce computational costs in simulation-intensive domains, and improve the reliability of high-resolution predictions in areas such as weather forecasting and turbulence modeling.

Despite these potential benefits, several limitations are relevant for deployment in high-stakes settings. Like most deep learning methods, IRNO lacks built-in uncertainty quantification, as its predictions do not come with calibrated confidence intervals, which limits their use in safety-critical decisions that require assessing prediction reliability. Additionally, the iterative refinement process introduces an additional computational overhead at inference time, and divergence is possible for large step sizes or out-of-distribution inputs. Extending IRNO with uncertainty quantification, for example, through ensemble or probabilistic methods, is a natural direction for future work that would broaden its applicability to decision-support contexts.

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

# A. Limitations and Future Work

The proposed Iterative Refinement Neural Operator (IRNO) demonstrates stable convergence, reduced spectral error, and improved cost–performance trade-offs across multiple physical systems. Several open questions remain that point toward productive directions for future research.

**Beyond Local Neighborhoods.** The present analysis establishes convergence under smoothness and spectral radius assumptions in a neighborhood of the true solution, depending on the quality of the initialization from the base operator. Extending these results to broader regions of the solution space remains an open problem. Tools from monotone operator theory, non-expansive mappings, or Lyapunov stability analysis may provide a path toward characterizing global behavior and basin structure for learned refinement dynamics.

**Adaptive Control of Refinement Dynamics.** IRNO uses a fixed step size chosen empirically to satisfy a contraction condition derived from the local Jacobian. This choice offers a balance between stability and convergence speed, though it does not incorporate information from the refinement trajectory itself. A natural criterion for adaptive stopping is when $\|\Phi_\theta(x, h_k)\|$ falls below a threshold tied to the bias level, which avoids over-refinement once the error floor has been reached. Future work may explore adaptive step-size policies or learned controllers that respond to local error geometry or spectral content, thereby drawing connections to classical line-search and trust-region methods.

**Extensions to Stochastic and Inverse Settings.** This work focuses on deterministic forward prediction tasks. Extending iterative refinement to stochastic operators, uncertainty-aware prediction, or inverse problems, such as data assimilation or parameter identification, represents a promising direction. In these settings, refinement dynamics could be coupled with probabilistic inference mechanisms, potentially bridging learned solvers with Bayesian filtering or variational frameworks.

**Cross-Resolution and Distribution Generalization.** Current evaluation uses matched train and test resolutions, and all benchmarks draw from the same data distribution. Whether the learned update rule transfers to finer grids than those seen during training, or remains accurate under distributional shift (e.g., out-of-distribution forcing terms or boundary conditions), is an open question. Studying resolution generalization and distributional robustness is a key direction for establishing the broader applicability of IRNO in scientific workflows.

# B. Related Work

### B.1. Spectral Bias and Multi-Scale Operator Learning

A fundamental challenge in training neural operators for SciML is the phenomenon of spectral bias, where neural networks prioritize learning low-frequency components of the target function while struggling to capture high-frequency details (Rahaman et al., 2019). Early efforts to address this limitation in the context of operator learning focused on architectural innovations that explicitly route information across scales. For instance, the Hierarchical Attention Neural Operator (HANO) (Liu et al., 2024) was proposed to enable nested feature computation that better captures multiscale solution spaces, by employing a scale-adaptive interaction range and self-attention mechanisms over a hierarchy of levels. More recent approaches have sought to mitigate spectral bias through frequency-specific boosting and scaling mechanisms. SpecBoost was introduced as an ensemble framework designed to enhance high-frequency capture in FNOs (Qin et al., 2024). Their spectral analysis revealed that FNOs exhibit a distinct parameterization bias that favors dominant low frequencies. To overcome this, SpecBoost trains a secondary residual operator specifically tasked with learning the high-frequency residuals left by the primary model. Similarly, High-Frequency Scaling (HFS) modulates the latent space of convolutional-based neural operators to amplify high-frequency modes (Khodakarami et al., 2026). Unlike Fourier-based interventions, HFS operates directly in the latent representation, avoiding the computational overhead of the FFT while successfully recovering fine-scale features in complex multiphase flow and turbulence problems.

The integration of generative models has emerged as a powerful strategy for recovering high-frequency spectral content, particularly in turbulent and multiscale systems. Conditioning diffusion models on the output of neural operators enables the generation of fine-scale fluctuations that are typically smoothed out by deterministic operator predictions, leading to improved alignment with the true energy spectrum (Oommen et al., 2025). From a resolution-generalization perspective, scale anchoring has been identified as a fundamental limitation in which models trained on low-resolution data fail to extrapolate to higher Nyquist frequencies, and Frequency Representation Learning has been proposed as a means to align spectral features across resolutions and reduce this dependency(Wang & Zhang, 2025). Our work complements these approaches but differs fundamentally in its inference mechanism. While these methods improve spectral coverage

through enhanced architectures or training objectives, they retain a *single-pass* inference structure where all frequency components must be resolved simultaneously. In contrast, IRNO reframes operator inference as an *iterative refinement process*, progressively correcting residual errors across scales through learned fixed-point iterations. This enables systematic error reduction at inference time without increasing the base model's capacity or retraining.

A concurrent line of work, F-Adapter (Zhang et al., 2026) (NeurIPS 2025), addresses spectral bias through parameter-efficient adaptation of Fourier layers using low-rank decompositions. Its theoretical grounding concerns the approximation capacity of LoRA-style adapters in the Fourier domain. IRNO's theoretical grounding is distinct, concerning the dynamics of fixed-point iteration and establishing contraction conditions for monotonic convergence. Empirically, F-Adapter targets minimal-parameter adaptation (2.31% VRMSE reduction on Active Matter) while IRNO trades additional training cost for substantially larger gains (50.73%), representing complementary design regimes rather than competing approaches. See Appendix H for a direct quantitative comparison.

### B.2. Numerical Principles as Inductive Bias

Deep learning architectures often exhibit structural similarity with classical numerical methods. Residual networks, for example, are mathematically equivalent to the forward Euler discretization of continuous-time dynamical systems (He et al., 2016). Similarly, encoder-decoder structures with skip connections, such as U-Nets, mirror the restriction and prolongation operations in multigrid methods, enabling efficient error reduction across scales (Haber & Ruthotto, 2017; Wienands & Joppich, 2005). Neural operators such as FNO (Li et al., 2021) and DeepONet (Lu et al., 2021) parameterize mappings in spectral or basis-function spaces, consistent with classical spectral/pseudo-spectral methods. In most cases, these models are trained as *direct solvers* that approximate the solution operator through a single forward evaluation.

A closer conceptual link to our framework arises in implicit layers, Deep Equilibrium Models (DEQs)(Bai et al., 2019), and Neural Ordinary Differential Equations (Neural ODEs)(Chen et al., 2018), which explicitly cast network inference as the solution of a fixed-point or continuous-time dynamical system. These approaches define the *network itself* as an equilibrium or flow, with iterative solvers embedded into training and inference. IRNO differs by preserving a standard, explicit base operator and introducing a learned *refinement dynamics* applied at inference time. Opposed to learning an implicit representation, IRNO learns a residual update rule that iteratively corrects the output of a pre-trained operator, retaining architectural modularity while inheriting the convergence and stability properties of classical fixed-point iterations.

## C. Proof of Theorem 3.1 (Quadratic–Linear Convergence)

We sketch the main steps establishing the error recursion and local contractivity.

Let $e_k = y - h_k$. From the iteration

$$h_{k+1} = h_k + \alpha \Phi(x, h_k),$$

and the local affine decomposition in Assumption 1,

$$\Phi(x, h_k) = A(x, h_k)e_k + b + R(x, h_k),$$

we obtain

$$e_{k+1} = e_k - \alpha A(x, h_k)e_k - \alpha b - \alpha R(x, h_k).$$

Decomposing $A(x, h_k) = A(x) + \big(A(x, h_k) - A(x)\big)$ yields

$$e_{k+1} = (I - \alpha A(x))e_k - \alpha\big(A(x, h_k) - A(x)\big)e_k - \alpha b - \alpha R(x, h_k).$$

Taking norms and applying the Jacobian stability and remainder bounds from Assumptions 1 and 2, we obtain

$$\|e_{k+1}\| \leq \|I - \alpha A(x)\|_{\text{op}}\|e_k\| + \alpha\mu\|e_k\|^2 + \alpha\frac{L}{2}\|e_k\|^2 + \alpha\|b\|.$$

Defining $q = \|I - \alpha A(x)\|_{\text{op}}$ and $c = \alpha\left(\frac{L}{2} + \mu\right)$ gives the claimed bound

$$\|e_{k+1}\| \leq q\|e_k\| + c\|e_k\|^2 + \alpha\|b\|.$$

When $b = 0$ and $\|e_0\| < \frac{1-q}{2c}$, we have

$$q + c\|e_k\| < 1,$$

which implies local contractivity. By induction, the error decreases monotonically and the iteration converges to $y$. The resulting behavior is linear for small errors, with a quadratic correction term governing the transient regime.

$\square$

The error recursion in Theorem 3.1 decomposes the refinement dynamics into three contributions with clear numerical interpretations. The linear term $q\|e_k\|$ governs the convergence rate and is determined by the choice of step size $\alpha$ and the spectral properties of the local Jacobian through $q = \|I - \alpha A(x)\|_{\text{op}}$. The quadratic term $c\|e_k\|^2$ captures higher-order deviations from the local linearization and dominates in the transient regime, explaining the accelerated error reduction observed in early refinement steps. Finally, the bias term $\alpha\|b\|$ sets a limiting error floor in the presence of model mismatch, leading to convergence toward a neighborhood of the true solution as formalized in Corollary 3.3.

# D. Proofs of Corollaries

**Corollary 3.2 Geometric Convergence and Iteration Complexity**

From Theorem 3.1 with $b = 0$, the error satisfies

$$\|e_{k+1}\| \le q\|e_k\| + c\|e_k\|^2.$$

As $\|e_k\| \to 0$, the quadratic term becomes negligible. For any $\varepsilon > 0$, there exists $k_0$ such that for all $k \ge k_0$,

$$c\|e_k\| < \varepsilon,$$

and hence

$$\|e_{k+1}\| \le (q + \varepsilon)\|e_k\|.$$

Iterating this inequality yields

$$\|e_k\| \le C(q + \varepsilon)^k \|e_0\|,$$

for some constant $C > 0$. Since $\varepsilon$ is arbitrary, this implies the asymptotic geometric rate $\|e_k\| \lesssim q^k \|e_0\|$.

To achieve $\|e_k\| \le \varepsilon'$, it suffices to choose $k$ such that

$$q^k \|e_0\| \le \varepsilon',$$

which gives

$$k = O\left(\frac{\log(\|e_0\|/\varepsilon')}{\log(1/q)}\right).$$

$\square$

**Corollary 3.3 Convergence with Bias**

**Part 1: Existence and Uniqueness of Fixed Point, and Convergence.**

Define the operator $T : B_\delta(y) \to \mathcal{H}$ by

$$T(h) = h + \alpha\Phi(x, h).$$

We now verify explicitly that $T$ maps a ball around $y$ into itself and is a contraction, so the Banach Fixed Point Theorem applies. The derivation also reveals how the admissible radius depends on $q$ and the bias $b$.

Let a positive $r < \delta$ be chosen later and consider $h \in \overline{B}_r(y)$. Using Assumption 1 gives

$$
\begin{aligned}
\|T(h) - y\| &= \|h - y + \alpha[b + A(x, h)(y - h) + R(x, h)]\| \\
&= \|(I - \alpha A(x, h))(h - y) + \alpha b + \alpha R(x, h)\|.
\end{aligned}
$$

Applying Assumptions 2 and triangular inequality, we have

$$
\begin{aligned}
\|T(h) - y\| &= \|(I - \alpha A(x,h))(h-y) + \alpha b + \alpha R(x,h)\| \\
&\leq \|I - \alpha A(x,h)\|_{\mathrm{op}} \|h-y\| + \alpha\|b\| + \alpha\|R(x,h)\| \\
&\leq \big(\|I - \alpha A(x)\|_{\mathrm{op}} + \alpha\|A(x,h) - A(x)\|_{\mathrm{op}}\big)\|h-y\| + \alpha\|b\| + \alpha\|R(x,h)\| \\
&\leq (q + \alpha\mu r)r + \alpha\|b\| + \alpha\frac{L}{2}r^2 \\
&= qr + \alpha\left(\mu + \frac{L}{2}\right)r^2 + \alpha\|b\|.
\end{aligned}
$$

Thus, $T$ maps $\overline{B}_r(y)$ into itself provided $r$ satisfies

$$
r \;\geq\; qr + \alpha\big(\mu + \tfrac{L}{2}\big)r^2 + \alpha\|b\|.
$$

This quadratic inequality in $r$ admits a positive solution whenever $q < 1$ and $\|b\|$ is sufficiently small to make the discriminant positive. In particular,

$$
r_-, r_+ = \frac{1 - q \pm \sqrt{(1-q)^2 - 4\alpha^2\big(\mu + \frac{L}{2}\big)\|b\|}}{2\alpha\big(\mu + \frac{L}{2}\big)},
$$

and operator $T$ is self-mapping if $r \in [r_-, r_+] \cup (0, \delta)$.

For small enough $\|b\|$, let $\varepsilon := \frac{4\alpha^2\big(\mu + \frac{L}{2}\big)\|b\|}{(1-q)^2}$. Then

$$
r_- = \frac{(1-q) - (1-q)\sqrt{1-\varepsilon}}{2\alpha\big(\mu + \frac{L}{2}\big)} = \frac{(1-q)\big(1 - 1 + \frac{1}{2}\varepsilon + O(\varepsilon^2)\big)}{2\alpha\big(\mu + \frac{L}{2}\big)} = \frac{\alpha\|b\|}{1-q} + O(\|b\|^2),
$$

by taking the first-order Taylor expansion of $\sqrt{1-\varepsilon}$.

For $h_1, h_2 \in \bar{B}_r(y)$, we bound the contraction factor of $T$ using the Fréchet differentiability of $\Phi$ and Assumption 1-2.

By the fundamental theorem of calculus for Fréchet derivatives,

$$
\Phi(x, h_1) - \Phi(x, h_2) = \int_0^1 D_h\Phi\big(x, h_2 + t(h_1 - h_2)\big)(h_1 - h_2)\, dt.
$$

Under Assumption 1, the derivative $D_h\Phi(x,h)$ exists and satisfies

$$
D_h\Phi(x,h) = -A(x,h) + E(x,h),
$$

where the remainder term $E(x,h)$ arises from the quadratic remainder $R$ and fulfills $\|E(x,h)\|_{\mathrm{op}} \leq C\|y - h\|$ for some $C > 0$ (since $R$ is quadratic in $\|e\|$).

Hence,

$$
\begin{aligned}
\|T(h_1) - T(h_2)\| &= \left\|(h_1 - h_2) + \alpha\int_0^1 D_h\Phi\big(x, h_2 + t(h_1 - h_2)\big)(h_1 - h_2)\, dt\right\| \\
&= \left\|\int_0^1 \Big[I + \alpha D_h\Phi\big(x, h_2 + t(h_1 - h_2)\big)\Big](h_1 - h_2)\, dt\right\| \\
&\leq \int_0^1 \Big\|I + \alpha D_h\Phi\big(x, h_2 + t(h_1 - h_2)\big)\Big\|_{\mathrm{op}}\, dt \;\cdot\; \|h_1 - h_2\|.
\end{aligned}
$$

Now, for any $\xi \in \bar{B}_r(y)$,

$$
\begin{aligned}
\big\|I + \alpha D_h\Phi(x,\xi)\big\|_{\mathrm{op}} &= \big\|I - \alpha A(x,\xi) + \alpha E(x,\xi)\big\|_{\mathrm{op}} \\
&\leq \|I - \alpha A(x,\xi)\|_{\mathrm{op}} + \alpha\|E(x,\xi)\|_{\mathrm{op}}.
\end{aligned}
$$

Using Assumption 2 and the bound on $E$,

$$\|I - \alpha A(x,\xi)\|_{\mathrm{op}} \leq \|I - \alpha A(x,y)\|_{\mathrm{op}} + \alpha \|A(x,\xi) - A(x,y)\|_{\mathrm{op}}$$
$$\leq q + \alpha\mu\|y - \xi\|,$$

and $\|E(x,\xi)\|_{\mathrm{op}} \leq C\|y - \xi\|$. Since $\|y - \xi\| \leq r$, we obtain

$$\left\|I + \alpha D_h \Phi(x,\xi)\right\|_{\mathrm{op}} \leq q + \alpha(\mu + C)r.$$

Choosing $r$ sufficiently small so that

$$\rho := q + \alpha(\mu + C)r < 1,$$

we have for all $h_1, h_2 \in \bar{B}_r(y)$

$$\|T(h_1) - T(h_2)\| \leq \rho\|h_1 - h_2\|,$$

which establishes that $T$ is a contraction on $\bar{B}_r(y)$.

**Conclusion.** Therefore, $T$ is a contraction mapping $\overline{B}_r(y)$ into itself. By the *Banach Fixed Point Theorem*, $T$ admits a unique fixed point $h^*$ in $\overline{B}_r(y)$, and the iteration $h_{k+1} = T(h_k)$ converges linearly to $h^*$ for any initial $h_0 \in B_\delta(y)$.

**Part 2: Limiting Error Bound.**

At the fixed point $h^*$, one can trivially conclude that

$$0 = \alpha\Phi(x,h^*) = \alpha\left(A(x,h^*)e^* + b + R(x,h^*)\right).$$

Rearranging gives

$$\alpha A(x,h^*)e^* = -\alpha b - \alpha R(x,h^*).$$

Taking norms on both sides and by Assumption 1,

$$\|\alpha A(x,h^*)e^*\| \leq \alpha\|b\| + \frac{\alpha L}{2}\|e^*\|^2.$$

Since

$$\|I - \alpha A(x,h^*)\|_{\mathrm{op}} \leq \|I - \alpha A(x)\|_{\mathrm{op}} + \alpha\|A(x,h^*) - A(x)\|_{\mathrm{op}} \leq q + \alpha\mu\|e^*\|,$$

if $q + \alpha\mu\|e^*\| < 1$, by the Neumann series, we have $\alpha A(x,h^*)$ invertible and with

$$\sigma_{\min}\left(\alpha A(x,h^*)\right) \geq 1 - \|I - \alpha A(x,h^*)\|_{\mathrm{op}} \geq 1 - q - \alpha\mu\|e^*\|.$$

Plugging back, we obtain

$$\|e^*\| \leq \frac{\alpha\|b\| + \frac{\alpha L}{2}\|e^*\|^2}{\sigma_{\min}\left(\alpha A(x,h^*)\right)} \leq \frac{a\|b\| + \frac{\alpha L}{2}\|e^*\|^2}{1 - q - \alpha\mu\|e^*\|}.$$

This is the same quadratic inequality we solved in proving self-mapping, and with first-order Taylor expansion, we have

$$\|e^*\| \leq \frac{\alpha\|b\|}{1 - q} + O(\|b\|^2).$$

$\square$

# E. Empirical Validation of Strong Monotonicity

Assumption 2 requires the linearization at the solution, $A(x,y)$, to be bounded and strongly monotone. Concretely, we assume there exist constants $0 < m \leq M < \infty$ such that

$$\langle A(x,y)e, e \rangle \geq m\|e\|^2, \qquad \|A(x,y)\|_{\mathrm{op}} \leq M,$$

for all $e \in \mathcal{H}$. A natural question is whether this local assumption is reasonable for a trained refinement network. Since strong monotonicity is not automatic for a learned neural network, we examine it empirically in a controlled 1-D synthetic setting and on the full U-Net models used in our main experiments.

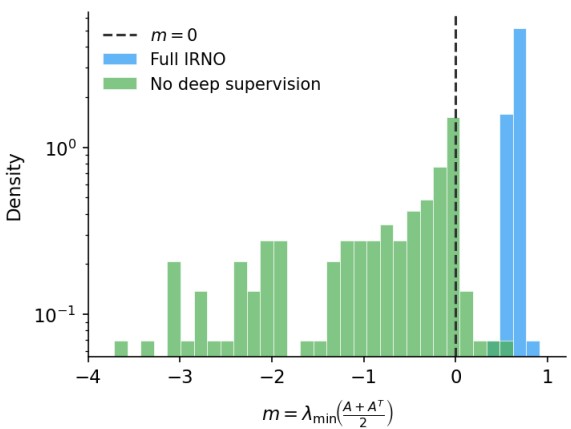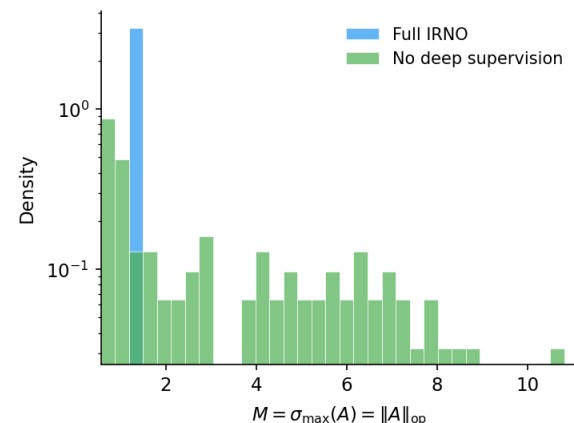

*Figure 9.* Distribution of the estimated strong-monotonicity constant $m = \lambda_{\min}((A + A^\top)/2)$ (left) and operator norm $M = \sigma_{\max}(A)$ (right) in the 1-D synthetic experiment, comparing Full IRNO and training without deep supervision. Full IRNO consistently yields positive $m$ and tightly bounded $M$, whereas removing deep supervision produces predominantly negative $m$ values and much larger variability in $M$. The dashed line in the left panel marks the boundary $m = 0$. Both y-axes are log-scaled.

**1-D synthetic experiment.** We consider the periodic problem

$$(I - \varepsilon L)y = \tanh(x),$$

with $n = 128$ and $\varepsilon = 0.3$, where $L$ is the discrete Laplacian with periodic boundary conditions and the solution is computed exactly by matrix inversion. We train a two-hidden-layer MLP refinement operator $\Phi_\theta$ with no architectural constraint enforcing monotonicity of the Jacobian. We compare two variants: **Full IRNO** and **No deep supervision**.

For each test pair $(x, y)$, we compute the exact Jacobian at the solution,

$$A(x, y) = -D_h \Phi_\theta(x, y),$$

using automatic differentiation, and measure

$$m = \lambda_{\min}\left(\frac{A + A^\top}{2}\right), \qquad M = \|A\|_{\text{op}} = \sigma_{\max}(A).$$

Since $\langle Ae, e \rangle = \left\langle \frac{A+A^\top}{2}e, e \right\rangle$ for all $e$, the quantity $m$ is the tightest strong-monotonicity constant.

Table 7 summarizes the results. Full IRNO yields uniformly positive $m$ and tightly concentrated operator norms across all test samples. In contrast, removing deep supervision leads to predominantly negative $m$ values and substantially larger, more variable operator norms, indicating that the bounded-and-strongly-monotone regime generally does not hold in that setting.

*Table 7.* Empirical monotonicity and boundedness of the learned Jacobian at the true solution in the 1-D synthetic experiment.

| Variant | $m$ (mean $\pm$ std) | $M$ (mean $\pm$ std) | $m > 0$ |
|---|---|---|---|
| Full IRNO | $0.661 \pm 0.060$ | $1.285 \pm 0.025$ | $100\%$ |
| No deep supervision | $-0.893 \pm 0.989$ | $3.087 \pm 2.571$ | $3.9\%$ |

Figure 9 visualizes the distributions of $m$ and $M$. The contrast between the two training procedures suggests that the proposed training objective, especially deep supervision, strongly promotes a locally monotone and well-conditioned refinement map near the solution.

**U-Net models on TR-2D and Active Matter.** We further evaluate the same criterion on the U-Net refinement models used in the main paper. Because exact Jacobian computation is expensive at this scale, we estimate the spectrum using power iteration. As shown in Table 8, the estimated monotonicity constant is positive on essentially all test samples, yielding

*Table 8.* Estimated strong monotonicity of the learned U-Net refinement operator on the main experimental benchmarks.

| Dataset | $m$ (mean $\pm$ std) | $m > 0$ |
|---|---|---|
| TR-2D U-Net (power iter.) | $13.71 \pm 2.41$ | $100\%$ |
| Active Matter U-Net (power iter.) | $3.01 \pm 3.12$ | $99.5\%$ |

$100\%$ on TR-2D and $99.5\%$ on Active Matter. These results support the relevance of the strong monotonicity assumption in realistic settings.

Overall, these experiments show that the bounded-and-strongly-monotone regime required by Assumption 2 emerges empirically under the proposed training procedure.

## F. Invariant-ball Condition

We derive conditions under which the ball $B_r(y)$ is forward invariant under the iteration $T(h) = h + \alpha\Phi(x, h)$. Let $h \in B_r(y)$ with $r \leq \delta$. Using the affine decomposition from Assumption 1,

$$
\begin{aligned}
\|T(h) - y\| &= \|h - y + \alpha\Phi(x, h)\| \\
&= \|(I - \alpha A(x, h))(h - y) + \alpha b + \alpha R(x, h)\| \\
&\leq \|I - \alpha A(x, h)\|_{\text{op}}\|e\| + \alpha\|b\| + \alpha\|R(x, h)\|.
\end{aligned}
$$

Applying Assumption 2 to bound $\|I - \alpha A(x, h)\|_{\text{op}}$,

$$
\|I - \alpha A(x, h)\|_{\text{op}} \leq \|I - \alpha A(x)\|_{\text{op}} + \alpha\|A(x, h) - A(x)\|_{\text{op}} \leq q + \alpha\mu r,
$$

and applying the remainder bound $\|R(x, h)\| \leq \frac{L}{2}\|e\|^2 \leq \frac{L}{2}r^2$ from Assumption 1, we obtain

$$
\|T(h) - y\| \leq qr + \alpha\mu r^2 + \alpha\frac{L}{2}r^2 + \alpha\|b\| = qr + cr^2 + \alpha\|b\|,
$$

where $c = \alpha\left(\frac{L}{2} + \mu\right) \geq 0$. For $B_r(y)$ to be forward invariant, we need $\|T(h) - y\| \leq r$, i.e.,

$$
qr + cr^2 + \alpha\|b\| \leq r \iff \alpha\|b\| \leq (1 - q)r - cr^2.
$$

**Case 1.** $c = 0$**.** The right-hand side is $(1 - q)r$, giving the condition

$$
\|b\| \leq \frac{(1 - q)r}{\alpha}.
$$

**Case 2.** $c > 0$**.** The function $(1 - q)r - cr^2$ is maximized at $r^* = \frac{1-q}{2c}$, yielding maximum value $\frac{(1-q)^2}{4c}$. Therefore, a valid $r \in (0, \delta]$ exists whenever

$$
\|b\| \leq \frac{(1 - q)^2}{4\alpha c},
$$

in which case the two positive roots of $(1 - q)r - cr^2 = \alpha\|b\|$ are

$$
r_\pm = \frac{(1 - q) \pm \sqrt{(1 - q)^2 - 4\alpha c\|b\|}}{2c},
$$

and $B_r(y)$ is forward invariant for any $r \in [r_-, r_+]$.

# G. Experimental Setup and Implementation Details

## G.1. Dataset Specifications

### G.1.1. TURBULENT RADIATIVE LAYER-2D (TR-2D)

**Data Generation.** The TR-2D dataset simulates Kelvin–Helmholtz instability using the ATHENA++ hydrodynamics code (Fielding et al., 2020). Initial conditions consist of two gas layers with different temperatures and velocities, creating shear-driven turbulence.

**Physical system.** This dataset models the interaction between hot, dilute gas and cold, dense gas moving at subsonic velocities, a configuration unstable to the Kelvin-Helmholtz instability. The turbulent mixing populates intermediate-temperature gas, which rapidly cools as heating and cooling become imbalanced. This process is fundamental to understanding phase structure in the interstellar and circumgalactic medium. The governing equations are given by

$$\frac{\partial \rho}{\partial t} + \nabla \cdot (\rho \vec{v}) = 0,$$

$$\frac{\partial (\rho \vec{v})}{\partial t} + \nabla \cdot (\rho \vec{v}\vec{v}) = -\nabla P,$$

$$\frac{\partial E}{\partial t} + \nabla \cdot ((E + P)\vec{v}) = -\frac{E}{t_{\text{cool}}},$$

where $\rho$ is density, $\vec{v}$ is the 2D velocity field, $P$ is pressure, $E$ is total energy, and $t_{\text{cool}}$ is the cooling time parameter.

**Dataset details.** The dataset contains 90 trajectories (10 random initializations for each of 9 cooling time values: $t_{\text{cool}} \in \{0.03, 0.06, 0.1, 0.18, 0.32, 0.56, 1.00, 1.78, 3.16\}$). Each trajectory consists of 101 timesteps at resolution 384×128, with spatial domain $x \in [-0.5, 0.5]$, $y \in [-1, 2]$ and temporal snapshots separated by $\Delta t = 1.597$ in simulation time. Fields include density, pressure, and velocity components.

**Task.** Given 4 consecutive frames $\{s_{t-3}, s_{t-2}, s_{t-1}, s_t\}$ where $s_t = (\rho_t, \vec{v}_t, P_t)$, predict the next frame $s_{t+1}$. This conditional prediction task tests the model's ability to capture short-term dynamics and turbulent evolution.

### G.1.2. ACTIVE MATTER (AM)

**Physical system.** This dataset simulates continuum dynamics of rod-like active particles immersed in a Stokes fluid. The system captures collective behavior, including energy transfer across scales, vorticity-orientation coupling, and phase transitions from isotropic to nematic states. The governing equations (Eqs. 1-5 in the associated paper) describe the evolution of concentration, velocity, orientation tensor, and strain-rate tensor fields.

**Dataset details.** The dataset contains 225 simulations spanning parameter space $\alpha \in \{-1, -2, -3, -4, -5\}$ (dipole strength), $\beta = 0.8$, and $\zeta \in \{1, 3, 5, 7, 9, 11, 13, 15, 17\}$ (alignment strength), with 5 trajectories per parameter set using different random initializations. Each trajectory consists of 81 timesteps at resolution 256×256, covering temporal range $t \in [0, 20]$ seconds with snapshots every 0.25 seconds. The spatial domain is $L_x = L_y = 10$ with periodic boundary conditions.

**Task.** Given 4 consecutive frames $\{s_{t-3}, s_{t-2}, s_{t-1}, s_t\}$ where $s_t = (\rho_t, \vec{v}_t, D_t, U_t)$ includes concentration, velocity, orientation tensor, and strain-rate tensor, predict the next frame $s_{t+1}$.

### G.1.3. ERA5 WEATHER SUPER-RESOLUTION

We conduct our experiments on the climate super-resolution benchmark provided by SuperBench (Ren et al., 2025), which is constructed from the ERA5 reanalysis dataset (Hersbach et al., 2020). In this work, we strictly follow the dataset construction and experimental protocol defined in SuperBench.

**Data Source and Task Definition.** ERA5 is a global atmospheric reanalysis dataset produced by the European Centre for Medium-Range Weather Forecasts (ECMWF) on a regular latitude–longitude grid. Following SuperBench, we focus on three climate variables: kinetic energy, temperature, and total column water vapor. We follow Scenario (i) (the general computer

vision setting) from SuperBench, in which low-resolution (LR) atmospheric states are obtained by bicubic downsampling without noise. The task is formulated as supervised spatial super-resolution, mapping LR climate fields of resolution $45 \times 90$ to corresponding high-resolution (HR) targets of resolution $720 \times 1440$, resulting in a $\times 16$ super-resolution factor along each spatial dimension.

**Dataset construction.** We follow the official data construction and temporal splits provided by SuperBench. Training data are selected from the years 2008, 2010, 2011, and 2013. The validation sets for interpolation and extrapolation (look-back) evaluation are drawn from 2012 and 2007, respectively. The test sets for the corresponding interpolation and extrapolation tasks are from 2009 and 2014–2015, respectively. All methods are trained and evaluated on the same splits to ensure fair comparison. For visualizations in the main paper, we use samples from the interpolation test set (year 2009).

**Data Preprocessing.** Following SuperBench, we standardize each variable using statistics computed from the training split, and apply the same normalization to the validation and test sets. No additional data augmentation or task-specific preprocessing is performed.

### G.1.4. CE-GAUSS (IRREGULAR MESH)

**Physical system.** CE-Gauss is an unstructured-mesh benchmark from the RIGNO dataset (Mousavi et al., 2025). It simulates convection-diffusion dynamics on irregular Gaussian-distributed point clouds, capturing transport phenomena on non-uniform spatial domains that are not representable on regular grids. The dataset provides 4 physical variables per node.

**Dataset details.** The mesh contains 16,384 nodes with irregular spatial distribution. We follow the official train/validation/test splits provided with the RIGNO benchmark. The evaluation task is autoregressive rollout over 7 timesteps.

**Task.** Given the current state on the irregular mesh, predict the next timestep. IRNO is applied with $K = 4$ refinement steps and $\alpha = 0.3$, using RIGNO as the base operator. This experiment shows that the IRNO framework extends to graph-based operators on irregular meshes.

The refinement model for CE-Gauss shares the same RIGNO architecture as the frozen base operator. It takes an 8-channel input formed by concatenating the current estimate $\hat{h}_k$ with the frozen model's initial prediction $\hat{h}_0$, and outputs a 4-channel correction field. The update is applied in normalized solution space, making the step-size $\alpha$ dimensionless and consistent across output variables. Training minimizes a trajectory supervision loss averaged over $K$ unrolled refinement steps rather than only the final-step loss, which stabilizes the iterative dynamics. At inference, refinement is applied at every autoregressive step as zero-shot generalization; the model is trained only for single-step prediction.

### G.2. Architecture Details

#### G.2.1. BASE OPERATORS

*Table 9.* Hyperparameters of FNO and TFNO across different datasets.

| Dataset | Model | Modes | Blocks | Hidden Size |
|---|---|---|---|---|
| TR-2D | FNO | 16 | 4 | 128 |
| | TFNO | 16 | 4 | 128 |
| Active Matter | FNO | 16 | 4 | 128 |
| | TFNO | 16 | 4 | 128 |
| ERA5 | FNO | 12 | 4 | 64 |

**Fourier Neural Operator (FNO).** We adopt the FNO2D from the Well and SuperBench as the base operator, which implements the Fourier Neural Operator (Li et al., 2021). Table 9 summarizes the hyperparameters used for FNO and TFNO across all datasets.

**Tucker-Factorized Fourier Neural Operator (TFNO).** To improve parameter efficiency and generalization, we also employ the TFNO (Kovachki et al., 2023). TFNO replaces the dense spectral convolution layers in the standard FNO with

Tucker-factorized tensor contractions. This decomposition allows for significant compression of the network weights while preserving expressivity. As shown in Table 9, the TFNO follows a similar architectural configuration to the FNO (in terms of depth and channel width).

**Wide Activation Super-Resolution (WDSR).**    We adopt the WDSR-A architecture (Fan et al., 2018) as a CNN-based base operator for scientific super-resolution within the SuperBench framework. The model follows a head–body–tail design and emphasizes local feature extraction through wide-activation residual blocks. Hyperparameter details are provided in Tables 10.

- **Residual Blocks.** The network body consists of 18 lightweight residual blocks, each using an expansion ratio of 4 and residual scaling factor of 0.1. Each block contains two $3 \times 3$ convolutional layers with ReLU activation and weight normalization.

- **Upsampling and Skip Connection.** The output is produced by a tail convolution followed by pixel-shuffle upsampling. In parallel, a skip branch directly upsamples the input using a separate convolution and pixel-shuffle, and the two paths are summed to form the final output.

*Table 10.* Hyperparameters of the WDSR-A model across different datasets.

| Dataset | #ResBlocks | Width | Expansion | Res. Scale | Upscale |
|---------|-----------|-------|-----------|-----------|---------|
| ERA5 | 18 | 32 | 4 | 0.1 | 16 |

### G.2.2. REFINEMENT OPERATOR $\Phi_\theta$

The refinement operator is implemented as a lightweight U-Net with the following specifications:

**Encoder Path.**    The encoder receives the concatenation of the original input $x$ and the current estimate $h_k$, resulting in $2C$ input channels. It consists of 3 levels of downsampling. Each level contains a **convolutional block** followed by a $2 \times 2$ **max-pooling** layer. Each convolutional block is composed of two successive sequences of:

$$\text{Conv2d}(3 \times 3) \to \text{BatchNorm2d} \to \text{GELU}$$

The number of feature channels doubles at each level, starting from $C_{\text{base}}$ and reaching $8C_{\text{base}}$ before the bottleneck.

**Bottleneck.**    The bottleneck bridges the encoder and decoder at the lowest spatial resolution. It consists of a single convolutional block (two $3 \times 3$ convolutions with BatchNorm and GELU) that processes the features in a $2^{\text{depth}}C_{\text{base}}$-dimensional latent space.

**Decoder Path.**    The decoder is symmetric to the encoder and performs 3 levels of upsampling. At each level, the feature maps are first upsampled using **bilinear interpolation** (scale factor of 2) followed by a $3 \times 3$ convolution that halves the channel dimension. These features are then concatenated with the corresponding skip connections from the encoder. The merged features pass through a standard dual-convolutional block.

**Output Head & Initialization.**    A final $1 \times 1$ convolution maps the $C_{\text{base}}$ hidden channels back to the output dimension. To ensure the iterative process starts with stable, small-magnitude corrections, we apply **Xavier uniform initialization** to the weights with a gain of 0.1 and initialize all biases to zero.

**Padding.**    To respect the periodic nature of the scientific datasets used in our benchmark, all $3 \times 3$ convolutional layers in $\Phi_\theta$ utilize **circular padding** instead of zero padding, ensuring spatial continuity.

**Parameter Count.**    Table 11 reports the parameter counts (in millions) of all models across different datasets. A cross ($\times$) indicates that the corresponding model is not used for that dataset. For our proposed Iterative Refinement Neural Operator (IRNO), we report the cumulative number of parameters required for a complete inference cycle consisting of $K$ refinement iterations. The last three rows correspond to large monolithic baselines with different refinement models, which are introduced and discussed in the main paper 4.7.

*Table 11.* Model parameter counts (in millions) across different datasets.

| Model | TR-2D(M) | Active Matter(M) | ERA5(M) |
|---|---|---|---|
| FNO (base) | 19.0 | 19.0 | 4.8 |
| TFNO (base) | 19.3 | 19.3 | $\times$ |
| WDSR (base) | $\times$ | $\times$ | 1.6 |
| Refinement $\Phi_\theta$ | 8.6 | 8.7 | 2.1 |
| FNO + IRNO ($K = 6$) | 27.6 | 27.7 | 6.9 |
| FNO + SRCNN | $\times$ | $\times$ | 6.6 |
| FNO + UNet ($\times 15$) | $\times$ | $\times$ | 37.3 |
| FNO + FNO | $\times$ | $\times$ | 9.5 |

---

**Algorithm 1** Training Iterative Refinement Neural Operator (IRNO)

---

1: **Input:** Dataset $\mathcal{D} = \{(x_i, y_i)\}_{i=1}^{N}$, base operator $\mathcal{T}_{\text{base}}$, refinement steps $K$, step size $\alpha$, weights $\beta_{\text{sp}}, \beta_{\text{fp}}$
2: **Initialize:** Refinement parameters $\theta$
3: **for** epoch $= 1, \ldots, E$ **do**
4:     **for** batch $(x, y) \in \mathcal{D}$ **do**
5:         $h_0 \leftarrow \mathcal{T}_{\text{base}}(x)$ {Base prediction (No gradient)}
6:         $\mathcal{L}_{\text{accum}} \leftarrow 0$
7:         **for** $k = 1$ **to** $K$ **do**
8:             $h_k \leftarrow h_{k-1} + \alpha \cdot \Phi_\theta(x, h_{k-1})$ {Refinement step}
9:             $\mathcal{L}_k \leftarrow \mathcal{L}_{\text{spatial}}(h_k, y) + \beta_{\text{sp}} \cdot \mathcal{L}_{\text{spectral}}(h_k, y)$
10:            $\mathcal{L}_{\text{accum}} \leftarrow \mathcal{L}_{\text{accum}} + \mathcal{L}_k$
11:         **end for**
12:         $\mathcal{L}_{\text{fp}} \leftarrow \beta_{\text{fp}} \cdot \|\Phi_\theta(x, y)\|^2$ {Fixed-point regularization}
13:         $\mathcal{L}_{\text{total}} \leftarrow \frac{1}{K} \mathcal{L}_{\text{accum}} + \mathcal{L}_{\text{fp}}$
14:         Update $\theta$ by minimizing $\mathcal{L}_{\text{total}}$
15:     **end for**
16: **end for**

---

### G.3. Training Configuration

**Base Operator Training.** In our experiments, the base operators are initialized from a pretrained checkpoint. The base operators are kept frozen during all subsequent training and evaluation stages.

**Refinement Operator Training.** Algorithm 1 provides the complete training procedure for IRNO, detailing the progressive refinement steps, combined spatial-spectral losses, and fixed-point regularization.

The refinement operator is optimized using AdamW with an initial learning rate of $3 \times 10^{-4}$ and weight decay $1 \times 10^{-5}$. We employ a cosine learning rate scheduler with a minimum learning rate of $10^{-6}$. Training is conducted for 250 epochs with a per-GPU batch size of 16, and gradient norms are clipped to a maximum value of 1.0 to ensure stability. During training, the refinement operator is unrolled for $K = 6$ and $K = 4$ iterative refinement steps when used with the FNO and WDSR/TFNO base models, respectively, with a fixed step size $\alpha_{refine} = 0.25$ for ERA5 dataset and $\alpha_{refine} = 0.2$ for TR-2D and AM datasets. The training objective consists of the standard reconstruction loss augmented with a spectral loss term, as described below.

**Baseline implementations.** We reproduce HFS (Khodakarami et al., 2026) and HiNOTE (Luo et al., 2024) from their respective official code repositories, evaluated on the ERA5 $16\times$ super-resolution task under the same dataset split and evaluation protocol used for IRNO.

**Progressive Spectral Loss Schedule.** We incorporate a spectral loss to encourage accurate reconstruction across different frequency bands. Two distinct mechanisms control the spectral emphasis. The frequency exponent $\lambda_k$ determines the

per-step weighting of high-frequency components, and the spectral loss weight $\beta_{\text{spectral}}$ scales the overall spectral term in the training objective.

For the TR-2D and Active Matter datasets, the frequency exponent $\lambda_k$ increases linearly from $\lambda_{\text{start}} = 1.0$ to $\lambda_{\text{end}} = 2.0$ over the $K$ refinement steps within each forward pass. The spectral loss weight $\beta_{\text{spectral}}$ undergoes a linear warm-up over the first 5 training epochs before being held fixed for the remainder of training.

For the ERA5 dataset, we adopt the same progressive exponent schedule ($\lambda_k \in [1.0, 2.0]$ over refinement steps). Table 12 shows that the progressive schedule improves ACC from 0.875 to 0.892 and reduces RFNE from 0.235 to 0.214 relative to a fixed exponent ($\lambda_k = 1.5$), confirming that progressive spectral emphasis benefits the super-resolution task.

*Table 12.* ERA5 spectral loss schedule ablation (FNO base, $K = 6$). Progressive $\lambda_k \in [1.0, 2.0]$ vs. fixed $\lambda_k = 1.5$.

| Schedule | ACC ↑ | RFNE ↓ |
|---|---|---|
| Fixed $\lambda_k = 1.5$ | 0.875 | 0.235 |
| Progressive $\lambda_k \in [1.0, 2.0]$ | **0.892** | **0.214** |

**Training Compute Budget.**  Table 13 reports per-batch wall-clock timing for the IRNO refinement operator (U-Net backbone) on Active Matter at different unrolling horizons $K$. Training overhead scales sublinearly in total wall-clock time, with $K = 4$ incurring $2.69\times$ overhead relative to $K = 1$ and $K = 6$ incurring $4.14\times$, reflecting the amortization of optimizer and data-loading costs across longer unrolls.

*Table 13.* Per-batch training timing (ms) for IRNO on Active Matter at different unrolling horizons $K$. Measurements on a single NVIDIA RTX PRO 6000 Blackwell Max-Q GPU.

| $K$ | **U-Net fwd** | **Loss** | **Backward** | **Optimizer** | **Total / batch** |
|---|---|---|---|---|---|
| 1 | 14.52 ms | 0.14 ms | 45.32 ms | 1.27 ms | 101.63 ms ($1\times$) |
| 4 | 54.69 ms | 0.62 ms | 170.86 ms | 1.13 ms | 273.89 ms ($2.69\times$) |
| 6 | 81.71 ms | 0.86 ms | 285.90 ms | 1.14 ms | 420.93 ms ($4.14\times$) |

### G.4. Implementation Details

**Software and Libraries.**  All experiments are implemented in Python 3.10.16 using PyTorch 2.5.1 with CUDA 12.1. NumPy and Matplotlib are used for numerical processing and visualization, respectively. FLOPs are measured using `fvcore`.

**Hardware.**  All experiments were conducted on NVIDIA GPUs. Specifically, we used NVIDIA L40 GPUs with 48 GB memory and NVIDIA RTX PRO 6000 Blackwell GPUs with 96 GB memory.

### G.5. Evaluation Metrics

#### G.5.1. PRIMARY METRICS

**Variance-scaled Root Mean Squared Error (VRMSE).**  For TR-2D and Active Matter, we report the variance-scaled root mean squared error (VRMSE)(Ohana et al., 2024), defined as the square root of the mean squared error normalized by the variance of the ground-truth field. Specifically, given a prediction $u$ and reference $v$ with spatial mean $\bar{v}$, we compute

$$\text{VRMSE}(u, v) = \left( \frac{\langle |u - v|^2 \rangle}{\langle |v - \bar{v}|^2 \rangle + \varepsilon} \right)^{1/2},$$

where $\langle \cdot \rangle$ denotes averaging over all spatial locations and output channels, and $\varepsilon$ is a small constant for numerical stability.

**Anomaly Correlation Coefficient (ACC).**  For ERA5 dataset, we report the Anomaly Correlation Coefficient (ACC) to assess the spatial pattern similarity between predictions and ground truth. Given a predicted field $\hat{u}$ and reference field $u$, we

first compute their anomalies by removing the spatial mean:

$$\hat{u}' = \hat{u} - \overline{\hat{u}}, \qquad u' = u - \overline{u},$$

where $\overline{(\cdot)}$ denotes the spatial mean. The ACC is then defined as

$$\text{ACC} = \frac{\langle \hat{u}', u' \rangle}{\sqrt{\langle \hat{u}', \hat{u}' \rangle \langle u', u' \rangle}},$$

where $\langle \cdot, \cdot \rangle$ denotes the inner product over all spatial locations and channels. ACC measures the similarity of spatial anomaly patterns and is particularly suitable for evaluating geophysical and climate variables.

**Relative Frobenius Norm Error (RFNE).**   We evaluate reconstruction accuracy using the Relative Frobenius Norm Error (RFNE), which measures the normalized discrepancy between the predicted field $\hat{u}$ and the ground-truth field $u$. Specifically, RFNE is defined as

$$\text{RFNE} = \frac{\|\hat{u} - u\|_F}{\|u\|_F},$$

where $\|\cdot\|_F$ denotes the Frobenius norm over all spatial dimensions and channels. RFNE provides a scale-invariant measure of global reconstruction error and is widely adopted in scientific super-resolution benchmarks.

### G.6. Computational Cost Analysis

**FLOPs Calculation.**   We compute floating-point operation counts (FLOPs) using PyTorch's `FlopCountAnalysis` with custom operator handlers designed to match the actual execution graph of our models. FLOPs are estimated at the operator level using the tensor shapes observed during a forward pass. Reported values correspond to a single forward pass of the base operator network, and cumulative FLOPs across multiple refinement steps are obtained by linear accumulation.

**Memory Profiling.**   We measure peak GPU memory usage during inference using PyTorch's CUDA memory profiler. All measurements are conducted with models in evaluation mode and under `torch.no_grad()` to exclude gradient storage.

Peak GPU memory usage is measured during inference using PyTorch's CUDA memory profiler. For the base FNO model, peak memory is obtained from a single forward pass. For iterative models with refinement steps (IRNO), memory usage is profiled throughout the refinement process by querying the peak allocated memory after each refinement step, and the maximum value observed across all iterations is reported as the peak memory usage for a given number of steps $K$. Specifically, we record the maximum allocated CUDA memory via `torch.cuda.max_memory_allocated`.

### G.7. Monolithic Residual Correction Models

Table 14 summarizes the architectural designs of the large monolithic refinement model baselines, performing single-pass inference, see 4.7.

*Table 14.* Architectural configurations of monolithic refinement model baselines used for comparison.

| Component | UNet($\times$15) | FNO | SRCNN |
|---|---|---|---|
| Architecture Type | U-Net encoder–decoder | Fourier Neural Operator | CNN-based network |
| Depth / Blocks | 5 encoder–decoder levels | 4 Fourier blocks | 3 convolutional blocks |
| Channel Width | [64, 128, 256, 512, 1024] | 64 | 256 |

## H. Additional Tables and Visualization

**Numerical values for scatter plots.**   Figure 8 in the main text presents the trade-offs between accuracy and computational cost for all compared models. To provide a precise quantitative reference, Table 15 lists the corresponding numerical values. For the iterative refinement baseline (IRNO), we report cumulative FLOPs, as well as peak memory usage across the entire inference process.

*Table 15.* Detailed numerical results corresponding to the scatter plots in the main text. The table reports ACC, RFNE, peak memory (MB), and cumulative FLOPs (G) for the base model, IRNO iterations, and the three large monolithic refinement model baselines.

| Model / Step | ACC(%) | RFNE (%) | Peak Memory (MB) | FLOPs (G) |
|---|---|---|---|---|
| Base (FNO) | 75.23 | 32.47 | 32469.00 | 87.74 |
| IRNO (K=1) | 77.30 | 31.32 | 32503.58 | 289.26 |
| IRNO (K=2) | 79.21 | 30.18 | 32503.58 | 490.77 |
| IRNO (K=3) | 80.92 | 29.08 | 32503.58 | 692.28 |
| IRNO (K=4) | 82.38 | 28.05 | 32503.58 | 893.80 |
| IRNO (K=5) | 83.58 | 27.13 | 32503.58 | 1095.31 |
| IRNO (K=6) | 84.55 | 26.32 | 32503.58 | 1296.83 |
| IRNO (K=7) | 85.33 | 25.63 | 32503.58 | 1498.34 |
| IRNO (K=8) | 85.94 | 25.03 | 32503.58 | 1699.86 |
| IRNO (K=9) | 86.43 | 24.53 | 32503.58 | 1901.37 |
| IRNO (K=10) | 86.82 | 24.10 | 32503.58 | 2102.89 |
| IRNO (K=11) | 87.13 | 23.75 | 32503.58 | 2304.40 |
| IRNO (K=12) | 87.38 | 23.45 | 32503.58 | 2505.91 |
| FNO + UNet ($\times 15$) | 78.82 | 29.80 | 33262.86 | 1131.39 |
| FNO + FNO | 76.50 | 32.13 | 33372.85 | 175.72 |
| FNO + SRCNN | 71.42 | 35.44 | 33237.83 | 1935.33 |

**Inference Time per Step.** We report the wall-clock inference time per step for different refinement pipelines on a single GPU. All measurements are conducted on an NVIDIA RTX PRO 6000 Blackwell Max-Q Workstation Edition GPU. For each method, the inference time is measured over five independent runs, and we report the mean and standard deviation. The reported numbers correspond to the average wall-clock time of one forward step, measured in milliseconds (ms).

*Table 16.* Inference time per step (ms) on a single NVIDIA RTX PRO 6000 Blackwell Max-Q GPU. Results are reported as mean $\pm$ standard deviation over five runs.

| Model | Time per step (ms) |
|---|---|
| FNO + IRNO (ours) | **308.29 $\pm$ 30.07** |
| FNO + FNO | 435.94 $\pm$ 1.01 |
| FNO + SRCNN | 653.85 $\pm$ 3.04 |
| FNO + UNet ($\times 15$) | 769.24 $\pm$ 23.99 |

**Visualization for ERA5 with $8\times$ upscale factor.** Figure 10 presents qualitative results on ERA5 for Kinetic Energy and Temperature Field under the $8\times$ spatial downsampling. The visualizations demonstrate that the proposed iterative correction framework consistently improves predictions across different physical fields and upsampling factors.

Figure 11 shows the temporal evolution of anomaly correlation coefficient (ACC) on ERA5 for three different physical fields under the $8\times$ downsampling. Across all fields, increasing the number of refinement steps consistently improves forecasting accuracy compared to the base FNO. These trends indicate that the iterative refinement framework generalizes well across both different fields and lower upsampling factors.

**Visualization for ERA5 with $16\times$ upscale factor.** Figure 12 presents additional qualitative results on ERA5 for the Temperature Field under the $16\times$ spatial downsampling, complementing Figure 1 in the main paper, which reports results on the Kinetic Energy Field. Visualizations demonstrate that the proposed iterative correction framework generalizes consistently across different physical fields.

Figure 13 reports the temporal evolution of anomaly correlation coefficient (ACC) on ERA5 for three different physical fields under the $16\times$ downsampling setting. Across all fields, increasing the number of refinement steps consistently improves forecasting accuracy compared to the base FNO. These trends are consistent across different fields, suggesting that the performance improvements are not field-specific but arise from the proposed iterative correction framework.

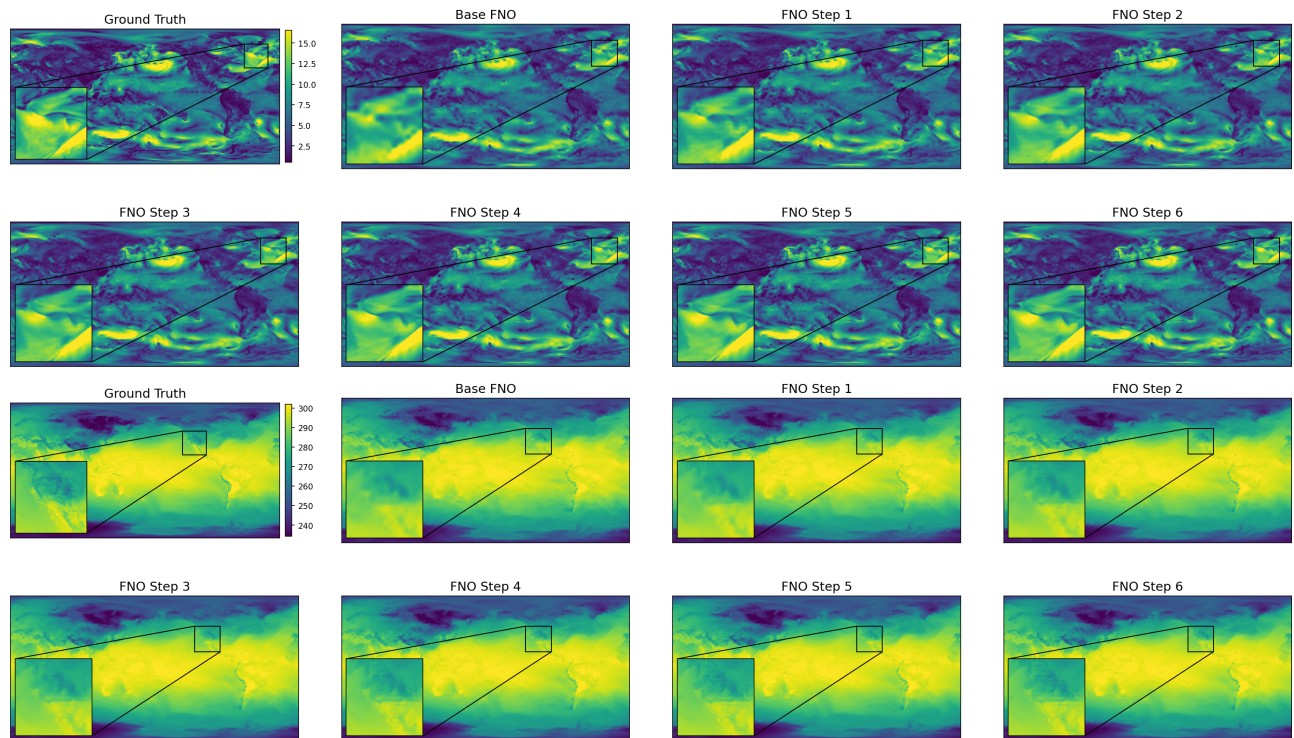

*Figure 10.* Qualitative results on ERA5 for Kinetic Energy (top) and Temperature Field (bottom) under $8\times$ spatial downsampling. The model is trained with up to **4 refinement steps**. We visualize the ground truth, the base FNO prediction, and the progressive refinement results after each refinement step.

Figure 14 illustrates the prediction results for three large monolithic baseline models. These single-step refinement (right column) still fail to resolve small-scale turbulent features. In contrast, our IRNO framework, see Figure 1 is more effective at capturing fine-grained physics.

**Visualization for TR-2D and Active Matter.** Figure 15 and 16 visualize predictions of FNO-based and TFNO-based IRNO on TR-2D density field. Figure 17 visualizes predictions of FNO-based and TFNO-based IRNO on Active Matter velocity field.

**Long-Horizon Extrapolation.** Figure 18 shows VRMSE as a function of refinement step $k$, evaluated beyond the training horizon of $K = 6$ up to $k = 48$ on Active Matter with FNO base operator. With step size $\alpha = 0.20$, the error reaches a minimum of approximately 0.0505 near $k \approx 14$ before diverging to 0.0806 at $k = 48$. With $\alpha = 0.05$, refinement remains stable throughout without divergence. Two complementary strategies can protect against divergence. First, step-size scheduling reduces $\alpha$ as $k$ increases. Second, adaptive stopping halts iteration when $\|\Phi_\theta(x, h_k)\|$ falls below a user-defined threshold tied to the bias level $\alpha\|b\|/(1 - q)$ from Corollary 3.3.

**HFS-ResUNet and IRNO Complementarity.** Table 17 reports VRMSE at each refinement step when IRNO is applied on top of an HFS-ResUNet base on Active Matter. Starting from the HFS base error of 0.0631, IRNO progressively reduces error to 0.0486 at $k = 6$, remaining stable at $k = 8$ (0.0487), demonstrating that iterative refinement compounds gains from frequency-aware architectures. Figure 19 further shows that IRNO achieves lower spectral error energy than HFS-ResUNet across the full radial frequency range.

*Table 17.* HFS-IRNO VRMSE per refinement step on Active Matter.

|  | **Base (HFS)** | $k = 1$ | $k = 2$ | $k = 3$ | $k = 4$ | $k = 6$ | $k = 8$ |
|---|---|---|---|---|---|---|---|
| VRMSE | 0.0631 | 0.0518 | 0.0504 | 0.0495 | 0.0490 | 0.0486 | 0.0487 |

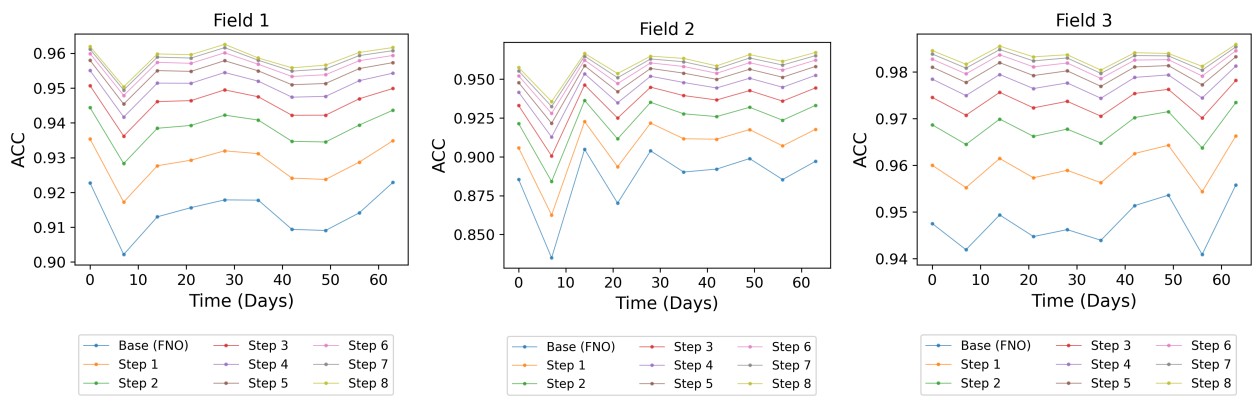

*Figure 11.* Temporal evolution of ACC on ERA5 under $8\times$ spatial downsampling for Field 1: Kinetic Energy, Field 2: Temperature, and Field 3: Precipitation (left to right). The model is trained with up to **4 refinement steps**. We report the base FNO and predictions obtained after different numbers of iterative refinement steps.

**Alternative Refinement Architectures.** Table 18 reports VRMSE for IRNO with different refinement backbone architectures on Active Matter (TFNO base, $K = 4$). All architectures achieve substantial error reduction, confirming that IRNO's gains are driven by the iterative mechanism rather than a specific backbone choice.

*Table 18.* IRNO with alternative refinement architectures on Active Matter (TFNO base, $K = 4$).

| Refinement Arch | Initial VRMSE | IRNO VRMSE | Reduction |
|---|---|---|---|
| ResNet | 0.1946 | 0.0430 | 77.88% |
| ConvNext | 0.1946 | 0.0438 | 77.43% |
| FNO | 0.1946 | 0.0559 | 71.26% |

**Normalization Ablation.** Table 19 compares IRNO trained with BatchNorm, LayerNorm, and GroupNorm on TR-2D (TFNO base, $K = 4$). All three normalization choices achieve consistent error reduction, confirming that the iterative refinement mechanism is robust to this architectural choice.

*Table 19.* Normalization ablation on TR-2D (TFNO base). VRMSE and improvement over base operator at $k = 4$ and $k = 8$.

| Normalization | VRMSE ($k = 4$) | Improv. ($k = 4$) | VRMSE ($k = 8$) | Improv. ($k = 8$) |
|---|---|---|---|---|
| BatchNorm (ours) | 0.1102 | 53.52% | 0.1042 | 56.05% |
| LayerNorm | 0.1038 | 56.61% | 0.0988 | 58.70% |
| GroupNorm | 0.1031 | 57.46% | 0.1006 | 58.50% |

**F-Adapter vs. IRNO.** Table 20 compares F-Adapter (Zhang et al., 2026) and IRNO (FNO base) on Active Matter. F-Adapter targets minimal-parameter adaptation, achieving $2.31\%$ VRMSE reduction with gains concentrated in mid- and high-frequency bands. IRNO trades additional training cost for substantially larger gains ($50.73\%$ overall) with broad spectral improvements across all frequency bands. Figure 20 visualizes the spectral error profile of F-Adapter and its per-frequency improvement over the FNO baseline.

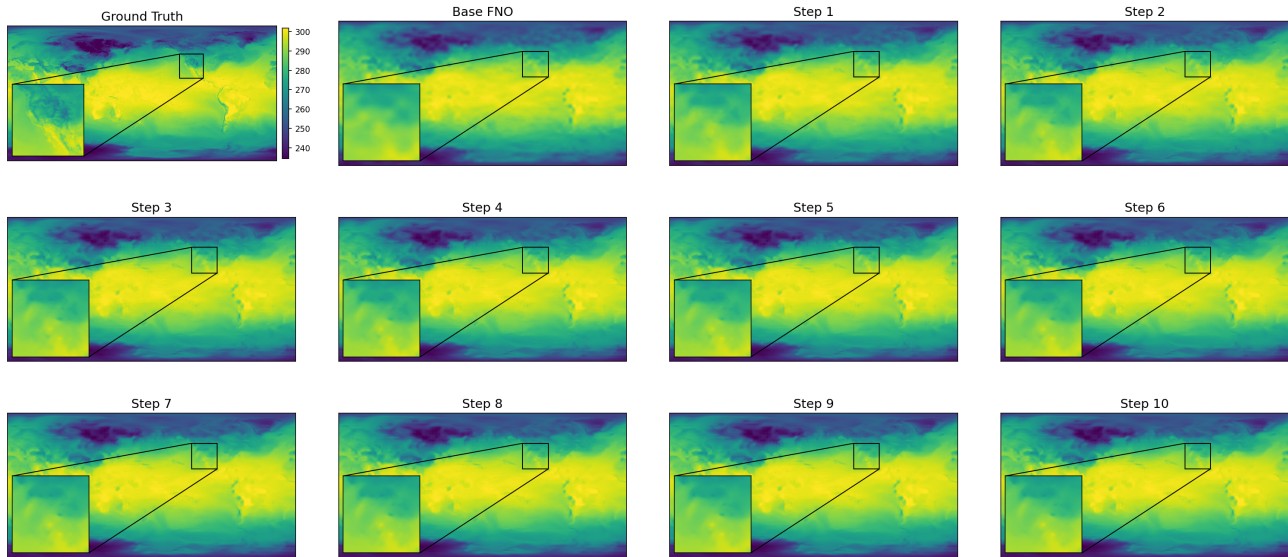

*Figure 12.* Qualitative results on ERA5 for Temperature Field under $16\times$ spatial downsampling. The model is trained with up to **6 refinement steps**. We visualize the ground truth, the base FNO prediction, and the progressive refinement results after each refinement step.

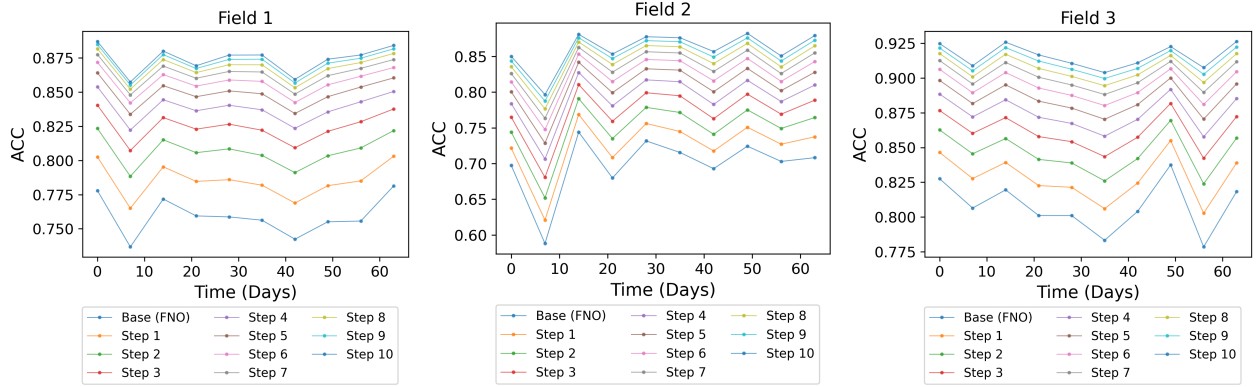

*Figure 13.* Temporal evolution of ACC on ERA5 under $16\times$ spatial downsampling for Field 1: Kinetic Energy, Field 2: Temperature, and Field 3: Precipitation (left to right). We report the base FNO and predictions obtained after different numbers of iterative refinement steps.

*Table 20.* F-Adapter vs. IRNO on Active Matter (FNO base). Frequency-band improvements are relative to the base FNO.

| Method | VRMSE | Improvement | Low freq | Mid freq | High freq |
|---|---|---|---|---|---|
| F-Adapter | 0.0994 | 2.31% | 6.8% | 58.6% | 79.6% |
| IRNO | 0.0501 | 50.73% | 58.4% | 67.8% | 92.3% |

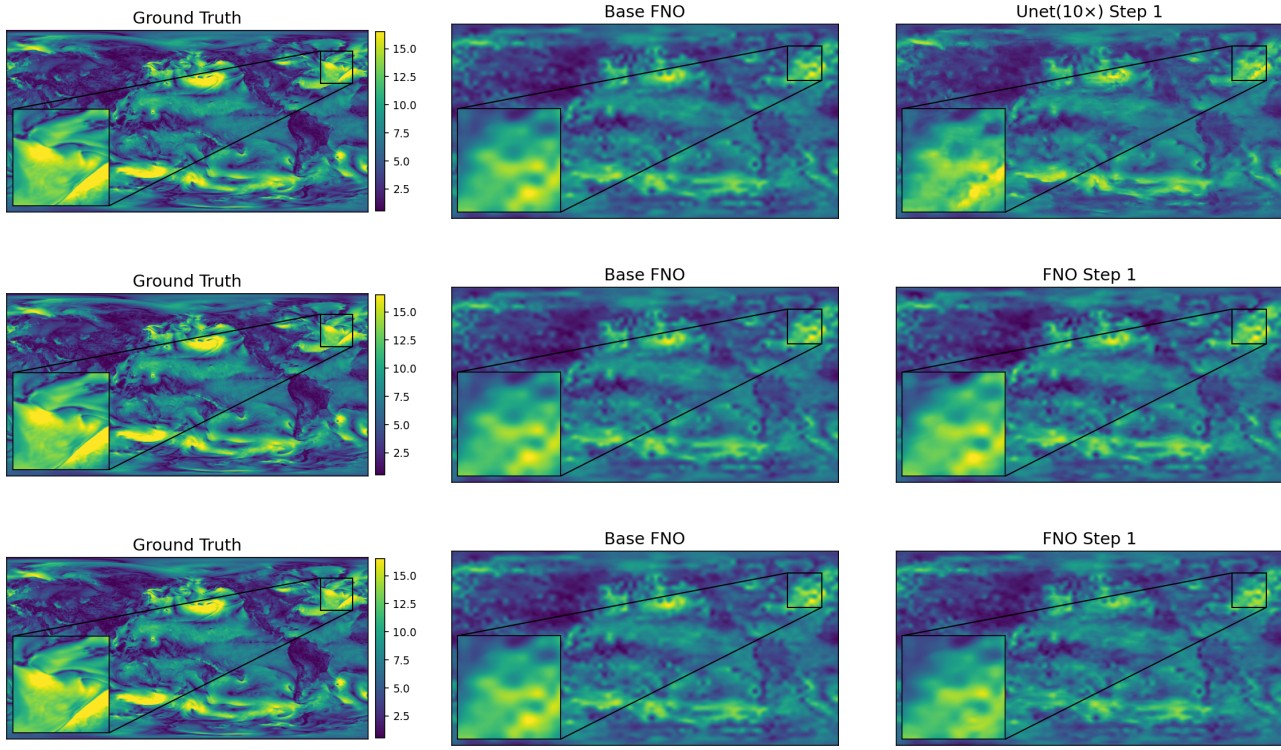

*Figure 14.* Qualitative comparison between Ground Truth, Base FNO, and single-step refinement across three large monolithic baseline architectures (Models UNet (15×), FNO, and SRCNN).

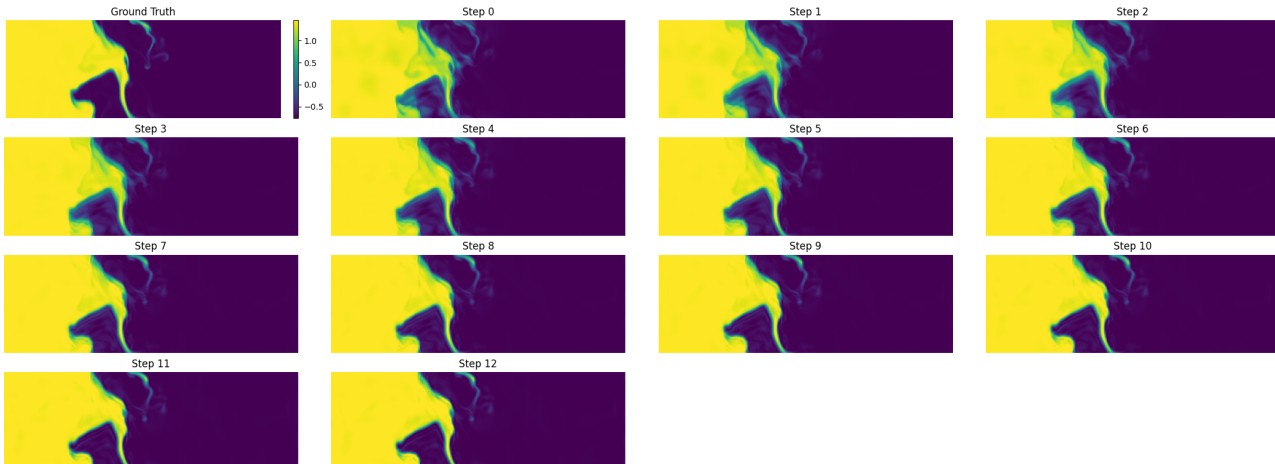

*Figure 15.* Qualitative results on TR-2D with FNO as base operator. The model is trained with up to **6 refinement steps**. We visualize the ground truth, the base FNO prediction, and the progressive refinement results after each refinement step.

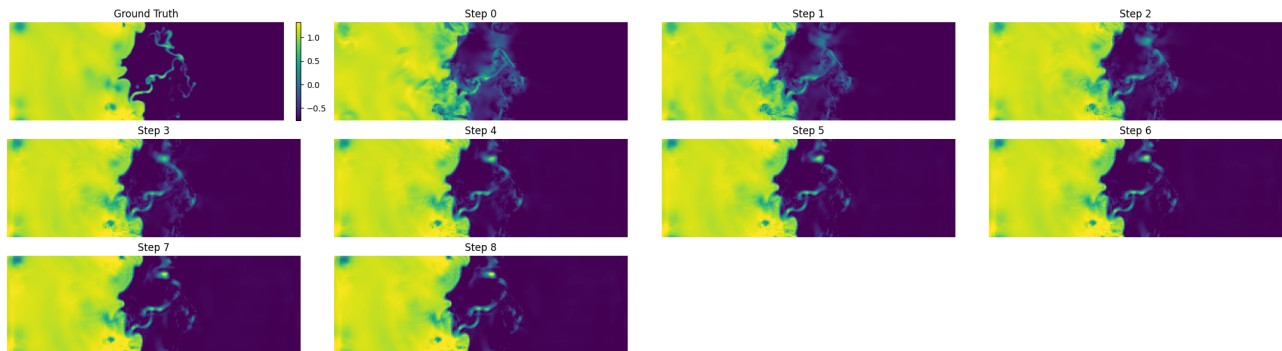

*Figure 16.* Qualitative results on TR-2D with TFNO as base operator. The model is trained with up to **4 refinement steps**. We visualize the ground truth, the base TFNO prediction, and the progressive refinement results after each refinement step.

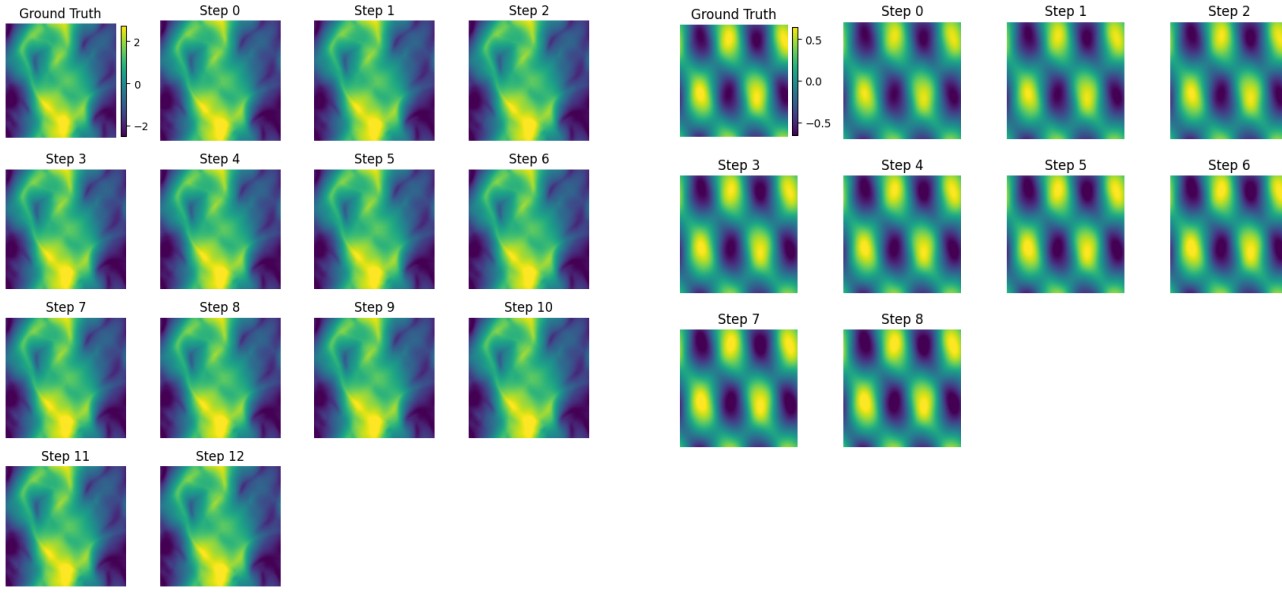

*(a)* FNO base operator (6 refinement steps)                *(b)* TFNO base operator (4 refinement steps)

*Figure 17.* Qualitative results on Active Matter with different base operators. We visualize the ground truth, base operator predictions, and progressive refinement results after each correction step.

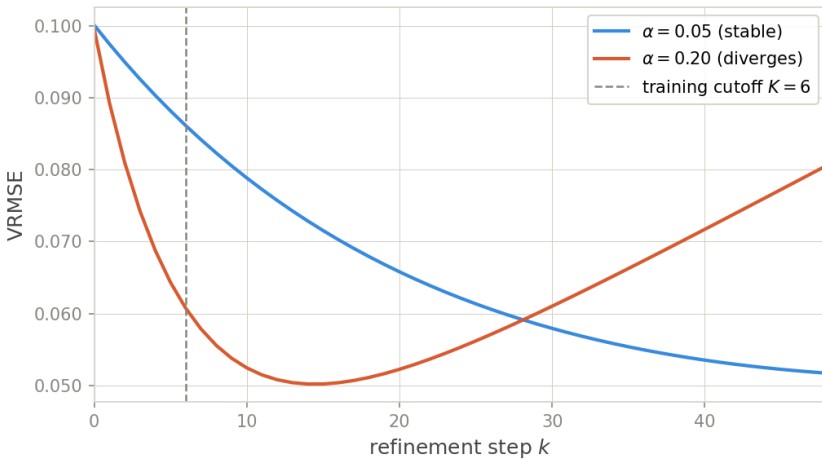

*Figure 18.* VRMSE vs. refinement step $k$ up to $8\times$ the training horizon ($K = 6$, dashed vertical line) for $\alpha = 0.05$ (stable) and $\alpha = 0.20$ (diverges) with FNO-based IRNO on Active Matter.

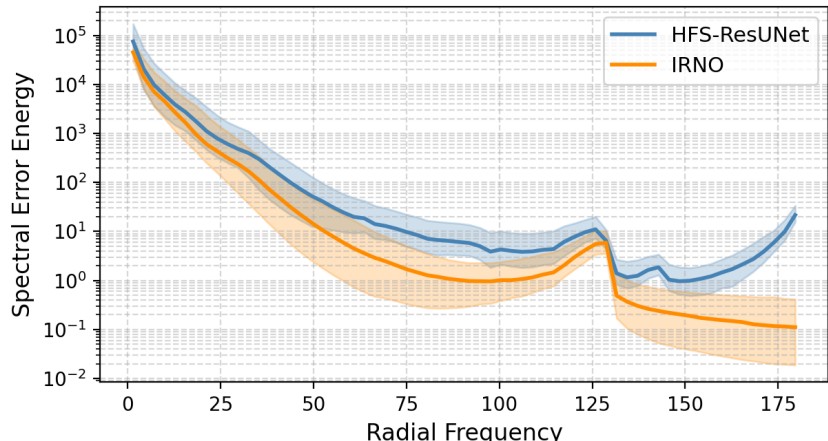

*Figure 19.* Spectral error energy vs. radial frequency for HFS-ResUNet and IRNO on Active Matter. Shaded regions denote $\pm 1$ standard deviation.

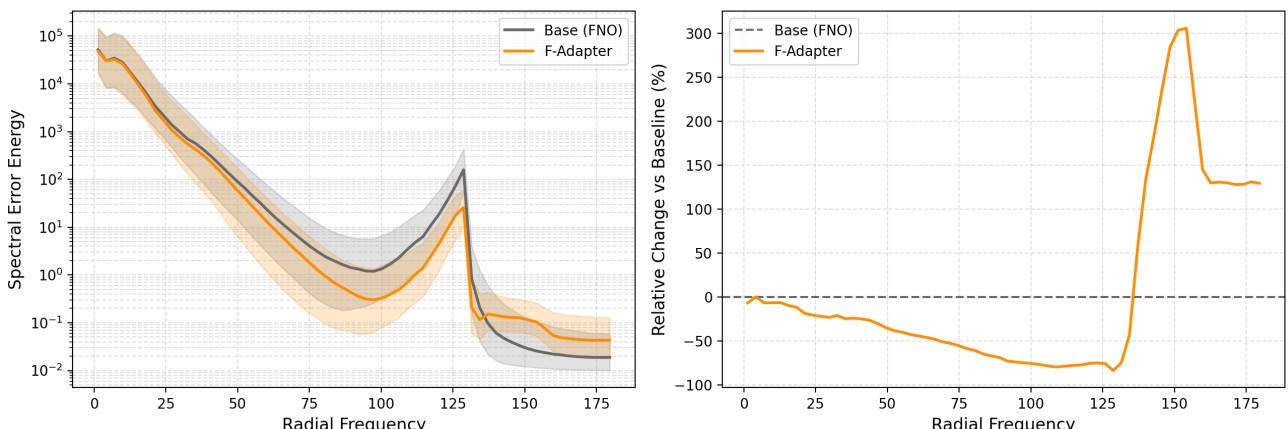

*Figure 20.* Spectral error vs. radial frequency for F-Adapter (left) and per-frequency improvement over FNO baseline (right) on Active Matter.

