# OpenReview forum: "Iterative Refinement Neural Operators are Learned Fixed-Point Solvers: A Principled Approach to Spectral Bias Mitigation"
_ICML.cc/2026/Conference — ICML 2026 spotlight_

### Official Review · Reviewer_rf1y · 2026-02-17

**Soundness:** 3
**Presentation:** 3
**Significance:** 3
**Originality:** 2
**Overall Recommendation:** 5
**Confidence:** 4

**Summary:**

This submission proposes shifting the paradigm of neural operators from a monolithic and single-pass inference procedure to a dynamic framework that parallels classical numerical solvers through iterative correction. The proposed Iterative Refinement Neural Operator (IRNO) utilizes a pre-trained base operator to provide a coarse initialization, and subsequently employs a shared-weight refinement module to resolve the prediction error via fixed-point iteration. To effectively train this system, the authors implement deep supervision across the trajectory and design a progressive spectral loss that adaptively increases penalty on high-frequency components during later steps to recover fine-scale details. Furthermore, the method incorporates fixed-point regularization to enforce the true solution as a fixed point and mitigate the residual error floor resulting from bias. Theoretically, the authors employ a local affine approximation to characterize the refinement process as a contraction mapping with guaranteed convergence to a unique fixed point. Empirical evaluations demonstrate the method's effectiveness.

**Compliance With Llm Reviewing Policy:**

Affirmed.

**Final Justification:**

Every question I raised has been completely resolved.

**Key Questions For Authors:**

1.  The paper claims the method is "architecture-agnostic" and that the refinement operator $\Phi_\theta$ can theoretically wrap any network. However, the current implementation relies on the concatenation of the input $x$ and the current estimate $h_k$ for U-Net style convolutional updates. This approach naturally fits grid-based outputs like those in FNO and TFNO. It is not immediately clear how this is plug-and-play for architectures like DeepONet [5] or methods dealing with non-Cartesian grids such as RIGNO [6]. These approaches would likely require additional interpolation, coordinate encoding, or graph-based modifications. Could the authors provide experimental results or implementation examples using DeepONet or RIGNO to demonstrate the engineering constraints and feasibility for such non-grid architectures?

    [5] DeepONet: Learning nonlinear operators for identifying differential equations based on the universal approximation theorem of operators

    [6] RIGNO: A Graph-based Framework For Robust And Accurate Operator Learning For PDEs On Arbitrary Domains, NeurIPS 2025

2.  The paper currently demonstrates stability for a limited number of extrapolation steps beyond the training horizon. Could the authors provide results for significantly longer extrapolation (e.g., $4 \times K$ or $8 \times K$ steps) and report on potential risks such as oscillation, degradation, or divergence after passing the optimal point? Discussion on corresponding protection strategies for such cases would be valuable.

3.  There are several inconsistencies regarding the frequency penalty parameter. Section 2.2.2 states that $\lambda_k$ "increases linearly from $\lambda_{start}$ to $\lambda_{end}$". In contrast, Appendix E.3 mentions a "fixed frequency exponent $\alpha=1.5$" for ERA5. This notation $\alpha$ conflicts with the step size symbol $\alpha$ defined earlier. Furthermore, the ablation study in Table 4 uses a progressive schedule $\lambda: 1 \rightarrow 2$, yet the text implies ERA5 uses a fixed value. Please clarify which parameter setting and symbol are correct for each experiment.

4.  There is a contradiction between the text and the provided code regarding the model architecture. Appendix E.2.2 explicitly states that "The encoder... consists of 3 levels of downsampling" and the decoder "performs 3 levels of upsampling". However, the provided `config.yaml` (line 15) and `model.py` (line 26) set `depth=4`. This discrepancy fundamentally alters the parameter count and the receptive field of the refinement operator.

5.  There appears to be a hidden issue in the normalization logic of the spectral loss. The formula in Section 2.2.2 differs from the implementation in `losses.py` (lines 57-63). The code first divides weights by their mean, then takes `torch.mean` across all dimensions (including the frequency dimensions), and finally divides the result by $H \times W$ again. Since `torch.mean` already averages over the spatial-frequency dimensions, the additional division by $H \times W$ causes the loss magnitude to be significantly smaller than what the formula defines. While the hyperparameter $\beta_{spectral}$ might compensate for this, the mathematical formulation does not match the codebase.

**Limitations:**

yes

**Strengths And Weaknesses:**

**Strengths**

1. The proposed approach is technically sound and practical to implement. The formulation of using a shared-weight refinement module combined with fixed-point iteration creates a clear framework for both training and inference. Furthermore, the authors have conducted empirical verifications addressing the core risks associated with iterative methods, such as stability.
2. The paper presents a complete and logical narrative. It effectively progresses from the motivation of mitigating spectral bias to the design of the iterative refinement mechanism. It then provides a theoretical explanation involving contraction and the residual error floor and concludes with comprehensive experiments covering convergence, spectral evolution, transferability, and the cost-performance Pareto frontier.
3. Addressing spectral bias is indeed a critical pain point in neural operators. The proposed architecture-agnostic approach is highly practical. If this method can be further extended to cross-resolution or cross-distribution settings, its potential impact would be significant.

**Weaknesses**

1. Structurally, the proposed method is closely related to existing techniques such as shared-weight residual recurrence, learned iterative solvers, and unrolled optimization. The core contribution appears to be a systematic combination of iterative refinement with a progressive spectral curriculum, alongside the demonstration of cross-operator transferability and cost-performance trade-offs. This represents a combinatorial innovation rather than a fundamental architectural paradigm shift.

2.  The design of the progressive spectral loss focuses primarily on the amplitude spectrum. While this effectively drives the energy distribution closer to the ground truth, the amplitude spectrum is insensitive to phase information. Phase is critical for determining the alignment of local features, sharp interfaces, and fine textures. Consequently, the model might produce outputs that appear spectrally correct but suffer from spatial drift or hallucinated textures. The paper lacks a systematic analysis of whether this frequency-domain curriculum introduces structural artifacts or biases in the spatial domain.

3. The evaluation primarily compares the method against the original base operator and a capacity-matched single-shot residual network. However, the solution space for high-frequency fitting includes other state-of-the-art approaches such as hierarchical attention [1], High-Frequency Scaling [2], or diffusion-based post-processing [3]. The absence of a systematic comparison with these relevant methods makes it difficult to assess the standing of IRNO within the broader SOTA landscape.

4. The authors observe that an IRNO trained on a weaker base operator can improve a stronger base operator and attribute this to learning robust error-correction strategies. While plausible, a simpler explanation exists: the network may have learned a generic denoising or sharpening function that naturally performs better on cleaner inputs. Stronger evidence is required to prove that the model is truly learning the error dynamics of the underlying PDE. This could include testing consistency across different PDE families or performing empirical estimations of the Jacobian or spectral radius.

5. Although the paper highlights a better Pareto frontier at inference time, the IRNO framework introduces significant training overhead due to unrolling $K$ steps and applying multi-step supervision. The analysis lacks a total end-to-end budget comparison. It remains unclear if IRNO is more cost-effective than simply fine-tuning the base operator or training a more sophisticated single-pass multi-scale model with an equivalent total computational budget.

6. There is a significant contradiction regarding the progressive spectral loss. Section 2.2.2 describes a frequency exponent $\lambda_k$ that linearly increases with the refinement step $k$ (which matches the code logic for `progressive_spectral_loss`). However, Appendix E.3 describes a schedule where the spectral loss *weight* increases over training epochs, which is a completely different mechanism. Crucially, the experiments on ERA5 are explicitly described as using a fixed frequency exponent without any progressive schedule. This implies that one of the core methodological contributions was not utilized in the most challenging benchmark. Additionally, the ablation in Table 4 only validates the progressive schedule on the Active Matter dataset, limiting the support for its generalizability.

7.  The theoretical analysis in Section 3 assumes the refinement operator $\Phi_\theta$ admits a local affine approximation, implying smooth behavior. However, the code implementation uses `nn.BatchNorm2d` in every block. In a shared-weight iterative setting, the running mean and variance in BatchNorm accumulate statistics across all intermediate steps $h_k$. Since the distribution of $h_k$ changes as the iteration progresses towards the solution $y$, these statistics represent a mixture of heterogeneous distributions, creating a theoretical-practical disconnect. Furthermore, using these mixed statistics during extrapolation ($k > K$) is risky as the distribution of $h_k$ may drift away from the training regime. The paper provides no theoretical justification for this choice or ablation studies comparing it with LayerNorm or GroupNorm.

8. The paper overlooks F-Adapter [4], a highly relevant recent work. F-Adapter specifically addresses frequency-dimension design and analyzes the spectral error floor and theoretical lower bounds. This work should be cited and discussed, particularly in Section 3 where the "bias-error floor" relationship is derived.


[1] An Attention-Based Spatio-Temporal Neural Operator with Uncertainty Quantification for Dynamical Systems, NeurIPS 2025

[2] Mitigating Spectral Bias in Neural Operators via High-Frequency Scaling, Neural Networks, 2025

[3] Wavelet Diffusion Neural Operator, ICLR 2025

[4] F-Adapter: Frequency-Adaptive Parameter-Efficient Fine-Tuning in Scientific Machine Learning, NeurIPS 2025

---

> ### Author Rebuttal · Authors · 2026-03-31
>
> We thank reviewer rf1y for their detailed and constructive feedback, especially on experiments. Full supporting material is available at [link](https://anonymous.4open.science/r/ICML-IRNO-Rebuttal-84C3/README.md).
>
> ---
> **W1. Combinatorial innovation over paradigm shift.** We believe the principled integration of shared-weight recurrence with learned iterative solvers in the neural operator setting is non-trivial, as it requires convergence guarantees in function space and a training objective for progressive spectral error correction. Empirically, we show IRNO yields strong results with solver-like behaviors.
>
> **W2. Amplitude-based spectral loss insensitive to phase.** We agree Fourier-magnitude supervision alone may be insufficient, but IRNO's training objective includes a multi-step spatial supervision, which directly penalizes spatial drifts. We observe no artifacts as reflected by strong spatial metrics.
>
> **W3. Comparison with state-of-the-art baselines.** We have added comparisons against hierarchical attention(HiNOTE)[1] and High-Frequency Scaling(HFS) on ERA5:
> |Method|ACC$\uparrow$|RFNE$\downarrow$|
> |--|--:|--:|
> |HFS|$0.892$|$0.225$|
> |HiNOTE|$0.906$|$0.222$|
> |**IRNO(WDSR)**|**$0.910$**|**$0.195$**|
>
> IRNO is also complementary to architectural spectral methods: on Active Matter, combining IRNO with HFS further reduces VRMSE from **0.0631** to **0.0486**. See supplementary material.
>
> **W4. Cross-operator transfer does not identify mechanism.** We agree transferability alone does not establish the error-correction mechanism. Here we provide two supporting arguments: architecturally, the refinement map is conditioned on the input $x$, so the update depends on physical context rather than local denoising; empirically, its local Jacobian $A$ is strongly monotone ($m=\lambda_{\min}(\frac{A+A^T}{2}) > 0$) on nearly all tested samples (TR-2D: $m=13.71$, 100% positive; Active Matter: $m=3.01$, 99.5% positive), consistent with the iterative-solver interpretation.
>
> **W5&8. End-to-end compute budget trade-offs.** We agree inference-time Pareto gains alone do not prove end-to-end cost-effectiveness. Still, IRNO’s training cost grows **sublinearly** with $K$ in our measurements ($4.14\times$ at $K=6$), while delivering substantially larger gains. On Active Matter, F-Adapter improves VRMSE by **2.31%**, whereas IRNO improves it by **50.73%**. This comparison suggests F-Adapter prioritizes adaptation efficiency, whereas IRNO targets larger accuracy gains at relatively higher training cost.
>
> **W6 and Q3&5. Inconsistencies in progressive spectral loss.** Thanks for catching this inconsistency. Two quantities were wrongly conflated in the Appendix: $\lambda_k$ is the frequency exponent that varies across refinement steps, whereas $\beta_{\mathrm{spectral}}$ is the loss weight that varies during warm-up. We re-ran ERA5 with the progressive schedule on FNO-based IRNO:
> |Setting|ACC $\uparrow$|RFNE $\downarrow$|
> |--|--:|--:|
> |Fixed $\lambda=1.5$|$0.875$|$0.235$|
> |Progressive $\lambda_k\in[1.0,2.0]$|**$0.892$**|**$0.214$**|
>
> The redundant division is indeed present in the codebase but absorbed by $\beta_{\mathrm{spectral}}$, leaving training unaffected as the reviewer noted. We will correct both the manuscript description and the implementation discrepancy on spectral loss noted by the reviewer.
>
> **W7. BatchNorm mixes statistics across steps.** We re-ran TR-2D(TFNO) with alternative normalizations. IRNO remains robust to the normalization choice:
> |Normalization|VRMSE|
> |--|--:|
> |BatchNorm|$0.1042$|
> |LayerNorm|$0.0988$|
> |GroupNorm|$0.1006$|
>
> **Q1. Architecture-agnosticism and applicability beyond Cartesian grids.** For graph-based operators, refinement can be performed directly in node-feature space without interpolation. We validate this on RIGNO-12 with CE-Gauss, where IRNO reduces autoregressive $L^2$ error by **12.5%–21.3%** across rollout steps; on structured grids, multiple refinement backbones remain effective. See supplementary material.
>
> We agree DeepONet requires special adaptation because of its branch-trunk factorization.
>
> **Q2. Long-horizon extrapolation and divergence.** We extended extrapolation to $k=48$ ($8\times$) on TR-2D. With $\alpha=0.2$, the error decreases to **0.0505** around $k=14$ and then gradually rises to **0.0806**; with $\alpha=0.05$, refinement remains stable. This motivates two practical safeguards: step-size scheduling and adaptive stopping based on update magnitude $|\Phi_\theta(x,h_k)|$. See supplementary material.
>
> **Q4. U-Net depth discrepancy.** `depth=4` denotes four resolution levels corresponding to 3 down- and up-sampling operations, consistent with Appendix E.2.2.
>
> ---
> We hope our response has addressed your concerns. Should any questions remain, we are happy to discuss further, and we hope you will consider revising your assessment.
>
> [1] Hierarchical Neural Operator Transformer with Learnable Frequency-aware Loss Prior for Arbitrary-scale Super-resolution, ICML 2024

---

> > ### Author Rebuttal · Reviewer_rf1y · 2026-04-01
> >
> > Thank you for the comprehensive response and your hard work during the rebuttal phase. Every question I raised has been completely resolved. I have increased my score to 5. Good luck to you.

---

> > > ### Author Response · Authors · 2026-04-01
> > >
> > > We sincerely thank the reviewer for acknowledging our rebuttal and for kindly raising their score to 5. We truly appreciate their detailed and constructive feedbacks.

---

### Official Review · Reviewer_8anx · 2026-02-17

**Soundness:** 4
**Presentation:** 4
**Significance:** 3
**Originality:** 4
**Overall Recommendation:** 5
**Confidence:** 2

**Summary:**

This paper introduces the Iterative Refinement Neural Operator (IRNO), a method designed to address spectral bias in neural operators. Standard neural operators struggle to capture fine, high-frequency details in a single pass. To fix this, IRNO adds a learned refinement module that acts as a test-time inference loop to progressively correct residual errors.

**Compliance With Llm Reviewing Policy:**

Affirmed.

**Final Justification:**

Score maintained after review of authors' rebuttal.

**Key Questions For Authors:**

N/A

**Limitations:**

The authors did not consider prediction under uncertainty. The Impact Statement is very weak. Since FNO architectures are difficult to interpret, using these opaque models raises concerns for critical decision-making applications.

**Strengths And Weaknesses:**

This paper addresses an important shortcoming of FNO. The numerical results are compelling. I appreciated the theoretical results that complement the experimental benchmarks. The experimental benchmarks and baseline comparisons are excellent.

---

> ### Author Rebuttal · Authors · 2026-03-31
>
> We thank reviewer 8anx for the positive and thoughtful assessment of our work, and we especially appreciate the acknowledgement of both the experimental quality and theoretical analysis.
>
> We also appreciate the reviewer's comment on this limitation. We agree that uncertainty-aware prediction is an important consideration for high-stakes downstream applications, and we will clarify the potential risks of applying IRNO without calibrated uncertainty estimation, especially in critical decision-making settings. We will also strengthen our Impact Statement accordingly.

---

> > ### Author Rebuttal · Reviewer_8anx · 2026-04-01
> >
> > Acknowledged

---

> > > ### Author Response · Authors · 2026-04-01
> > >
> > > We sincerely thank the reviewer for their acknowledgement and positive assessment.

---

### Official Review · Reviewer_6eCN · 2026-03-11

**Soundness:** 2
**Presentation:** 4
**Significance:** 2
**Originality:** 2
**Overall Recommendation:** 4
**Confidence:** 4

**Summary:**

Authors propose to improve accuracy of trained neural operators by learned iterative correction inspired by fixed-point iterations. A neural network that performs correction is trained with special loss aiming to mitigate poor approximation of high frequencies known as f-principle or spectral bias.

The resulting framework is supported both by theoretical considerations and numerical results that include ablation study and monitoring for the reduction of error in frequency domain.

**Compliance With Llm Reviewing Policy:**

Affirmed.

**Final Justification:**

The authors successfully addressed my concerns in the rebuttal:
1. The theory is now in better shape, with several problematic parts patched by refined assumptions.
2. The experimental part now clearly points toward the benefits of iterative updates.

**Key Questions For Authors:**

**Problems with theory**

Theoretical analysis starts in Section 3 by a list of several assumptions, which I personally find unnatural. Since this is somewhat subjective I will focus mostly on technical problems.

1. (lines 199, 200\) A subsection is titled "Lipschitz continuity of Linear Operator and Spectral Radius". First, bounded (by assumption) linear operators are always Lipschitz continuous. Second, the Lipschitz continuity authors ask that it is not for linear operators, rather it is for an operator-valued map $\\mathcal{H}\\rightarrow \\mathcal{L}$. That is, we consider $A(x, h)$ as a function of $h$ for fixed $x$ and require this map to be Lipschitz continuous. On line 210 authors again use "The Lipschitz continuity of linear operator assumption". All that creates confusion. I suggest this part to be formulated more carefully.

2. (lines 207, 208\) Authors ask to select step size $\\alpha$ such that  $\left\\|I \- \\alpha A(x, y)\right\\|_{\\text{op}}\\leq q \< 1$. Now, this may well be impossible to do by selection of $\\alpha$ alone. For example, consider $\\mathcal{H} \= \\mathbb{R}^{n}$ with $L\_{2}$ norm. Linear operator $A$ in this space is matrix $\\mathbb{R}^{n\\times n}$. Let $A$ be symmetric matrix, so it has a form $A \= Q D Q^{T}$ for some orthogonal matrix $Q$ and diagonal matrix $D$. The condition above becomes $\\left\\|I \- \\alpha D\\right\\|\_2 \\leq q \< 1$. If $D$ has any negative number on the diagonal, the $L\_2$ norm is $\>1$ for any $\\alpha\\in (0, 1\]$. Similarly if symmetric matrix $A$ does not have full rank, i.e., $D$ has zero on the diagonal, $\\left\\|I \- \\alpha D\\right\\|\_2 \\geq 1$ for arbitrary $\\alpha$.

   More abstractly, authors forget to ensure their system is stable. A standard way to do that is to apply Lyapunov theory.

3. (lines 178, 179\) "Local Affine Approximation". Authors assume the condition on the refinement operator holds in some neighbourhood of fixed point. I will show that under assumptions the author made (and under additional assumptions on stability) one can construct an iterative map with iterations that escape from this neighbourhood. This creates an obvious problem: if the sequence can escape, conditions of the iterative map no longer hold and anything can happen.

   Consider $\\mathcal{H} \= \\mathbb{R}$, take $y \= 0$, consider fixed $x$ (omitted in the whole example). Let iterations  be $h\_{n+1} \= \\frac{1}{2}h\_{n} \+ \\alpha b$. These iteration complies with all assumptions: $A$ is a real number taken to be $A=\\frac{1}{2\\alpha}$, so stability is ensured and condition $\\left\\|I \- \\alpha A\\right\\|\\leq q \< 1$ hold for $q \= \\frac{1}{2}$. Term $B$ is completely absent, operator valued map is Lipschitz continuous. Clearly, the sequence $h\_{n+1} \= \\frac{1}{2}h\_{n} \+ \\alpha b$ converge from arbitrary starting point since $h\_1 \= \\frac{1}{2} h\_0 \+ \\alpha b$, $h\_2 \= \\frac{1}{2^{2}} h\_0 \+ (\\alpha b \+ \\frac{1}{2} \\alpha b), \\dots, h\_{\\infty} \= 0 \+ \\alpha b\\sum\_{i=0}^{\\infty}\\frac{1}{2^{i}} \= 2 \\alpha b$. Since we have no assumptions on $\\alpha$ and $b$, the fixed point can be arbitrary far away from old fixed point $y \= 0$. This mean sequence escape the neighbourhood of $y$, where nothing is knew (assumed) about the itrations.

   It seems this part may be patched by some assumption on the bias term, or, more explicitly, some assumption on where a new fixed point resides relative to the old fixed point $y$.

4. Even worse, I can construct divergent iteration. For that we consider extended example with nonzero operator $B$. Iteration reads $h\_{n+1} \= \\frac{1}{2}h\_{n} \+ \\alpha b \+ \\alpha \\beta h\_{n}^2$. Now, all conditions on $A$ hold for arbitrary $\\alpha$. Condition on $B$ also hold with $L \= 2 |\\beta|$. Since iteration are globally defined, $\\delta$ is arbitrary large. This gives us initialisation quality conditions $|h\_{0}| \< \\frac{1}{4\\alpha |\\beta|}$ because $\\mu$ is arbitrary small (operator valued map is constant). Lets select $\\beta \= \-\\frac{1}{16}$, $\\alpha \=  1$, $b \= \-1$. In this case we have $|h\_0| \< 4$ and iteration becomes $h\_{n+1} \= \\frac{1}{2}h\_n \- \\frac{1}{16}h\_n^2 \- 1 \= \-\\left(\\frac{1}{4}h\_n \- 1\\right)^2$. Suppose we start from $h\_0 \= \-1$. In this case $h\_1 \= \- \\left(-\\frac{1}{4}-1\\right)^2 \< h\_0 \\leq \-1$, $h\_2 \< h\_1 \\leq \-1$ and so the sequence diverge to $-\\infty$.

In my view theory is in poor shape and needs substantial refinement.

**Experimental results**

The proposed method consists of two parts:

1. Iterative update with shared parameters (same model is applied repeatedly)
2. Loss function with frequency-based reweighting.

To test whether the iterative update is necessary, authors use residual network as a baseline. Unfortunately, I can not find where this residual network is compared with the performance of the iterative scheme. I kindly ask the authors to point out where this information can be found, or provide it in the rebuttal.

The importance of the loss does not seem to be tested completely. Authors provide the study how the choice of exponent influences final error. However, a more interesting question is what will happen if one trains a residual model with the same frequency-weighted loss. It still may be the case that the main reason for improvement is not the iterative update, but the special loss mitigating spectral bias.

**Limitations:**

yes

**Strengths And Weaknesses:**

I find the text of the article easy to follow. In general, notation is well explained, illustrations provide useful examples, primal ideas make sense and experimental results are mostly pointing toward the applicability of approach. That being said, I still find theory problematic and experiments not entirely convincing.

The idea of learning a fixed-point iteration is appealing. Unfortunately it is not new and hardly attainable in the black-box learning purely from data. Naturally, when there are no assumptions about the system being learned, it is very hard to construct contraction mapping. The authors tried to substitute knowledge about the system by introducing some extra assumptions on the map being learned, but, in my view, failed to do this properly. Because of that, the theoretical part needs a thorough revision.

On the experimental side, in my view, the results do not firmly support the claim that the main reason for improvement is a fixed-point iterative update.

I make these points explicit in the next section of the review.

---

> ### Author Rebuttal · Authors · 2026-03-31
>
> We thank reviewer 6eCN for their detailed and constructive feedback, especially on theory. Full supporting material is available at [link](https://anonymous.4open.science/r/ICML-IRNO-Rebuttal-84C3/README.md).
>
> ---
> **W1. Misleading description of the Lipschitz assumption**
>
> We thank the reviewer for pointing this out. We will rename the subsection to **“Lipschitz continuity of the operator-valued map $h \mapsto A(x,h)$”** to make clear that the assumption is imposed on the map $h \mapsto A(x,h)$ for fixed $x$.
>
> ---
> **W2. Missing assumption for $A(x,y)$**
>
> We thank the reviewer for identifying this missing condition. Step-size selection alone does not ensure $\lVert I - \alpha A(x,y) \rVert_{\mathrm{op}} < 1$ in general. In the revision, we will explicitly assume that $A(x,y)$ is locally **bounded** and **strongly monotone**, i.e., $\langle A(x,y)e,e \rangle \ge m \lVert e \rVert^2, \lVert A(x,y) \rVert_{\mathrm{op}} \le M$ for some $0 < m \le M < \infty$.
>
> Under this condition, choosing $0 < \alpha < \frac{2m}{M^2}$ guarantees $\lVert I - \alpha A(x,y) \rVert_{\mathrm{op}} < 1$, since for any unit vector $e$, $\lVert (I - \alpha A)e \rVert^2 = 1 - 2\alpha \langle Ae,e \rangle + \alpha^2 \lVert Ae \rVert^2 \le 1 - 2\alpha m + \alpha^2 M^2 < 1.$
>
> Equivalently, with the Lyapunov function $V(e) = \lVert e \rVert^2$, the linearized dynamics satisfy $V_{k+1} \le (1 - 2\alpha m + \alpha^2 M^2) V_k,$ so the energy decreases strictly whenever $0 < \alpha < \frac{2m}{M^2}$.
>
> We also provide empirical support for this assumption. In a controlled 1D synthetic problem, **Full IRNO** training yields Jacobians whose symmetric part is uniformly positive and whose norm is tightly bounded, while removing deep spatial supervision destroys this behavior:
> |Variant|$m$ (mean $\pm$ std)|$M$ (mean $\pm$ std)|$m>0$|
> |--|--:|--: |--:|
> |Full IRNO|$0.661\pm0.060$|$1.285\pm0.025$|$100$%|
> |No deep supervision|$-0.893\pm0.989$|$3.087\pm2.571$|$3.9$%|
>
> We also include power-iteration estimates on trained U-Nets:
> |Dataset|$m$ (mean $\pm$ std)|$m>0$|
> |--|--: |--: |
> |TR-2D U-Net|$13.71\pm2.41$ |$100$%|
> |Active Matter U-Net|$3.01\pm3.12$|$99.5$%|
>
> We will incorporate this revised assumption, derivation, and empirical justification in the final manuscript.
>
> ---
> **W3&4. Escaping iterations without an invariant-ball condition**
>
> We agree that without a bound on the bias term $b = \Phi(x,y)$, the iterates may escape the local neighborhood and break the assumptions. In the revision, we will state the required condition for the *invariant-ball* explicitly.
>
> Let $T(h) = h + \alpha \Phi(x,h)$. From the proof of Corollary 3.3, for $h \in B_r(y)$, $\lVert T(h) - y \rVert \le qr + cr^2 + \alpha \lVert b \rVert,$ where $c = \alpha(\frac{L}{2} + \mu) \ge 0$. To ensure $B_r(y)$ is forward invariant, we require $qr + cr^2 + \alpha \lVert b \rVert \le r.$
>
> When $c > 0$, this is feasible only if $\lVert b \rVert \le \frac{(1-q)^2}{4\alpha c}$; when $c = 0$, the condition reduces to $\lVert b \rVert \le \frac{(1-q)r}{\alpha}$.
>
> These conditions explain the reviewer’s examples. In the linear case ($c = 0$), forward invariance requires $2\alpha \lVert b \rVert \le \delta$, which keeps the iterates inside the local ball $B_\delta(0)$. In the quadratic example, our condition gives $\lVert b \rVert \le 1$. The reviewer’s choice $b = -1$ lies exactly at this threshold, so the trajectory converges to the stable fixed point $h^* = -4$. If $\lVert b \rVert > 1$, divergence will indeed occur.
>
> We will add this condition explicitly and clarify its role in the theory.
>
> ---
> **5. Comparison with Residual Models Trained with Progressive Spectral Loss**
>
> The comparison of IRNO with residual network is shown in Figure 8 to demonstrate cost-performance trade-off. We agree that the original draft did not fully isolate the contribution of the iterative update from spectral loss. To address this, we trained two residual baselines with the progressive spectral loss: SpecBoost [1] and a ConvNext residual model.
>
> On Active Matter, both SpecBoost and ConvNext show limited improvement in normalized error ratios across frequency bands:
>
> | Frequency band | SpecBoost error ratios | ConvNext error ratios |
> | :--- | :--- | :--- |
> | Low | 0.8567 | 0.987 |
> | Mid | 1.1376 | 0.998 |
> | High | 1.0277 | 1.001 |
>
> SpecBoost's above $1$ mid- and high-frequency error ratios are consistent with FNO's architectural low-frequency bias. Together, these two baselines show that progressive spectral reweighting alone does not produce the systematic spectral bias mitigation that IRNO achieves.
>
> ---
>
> We hope our response has addressed your concerns. Should any questions remain, we are happy to discuss further, and we hope you will consider revising your assessment.
>
> [1] Toward a better understanding of fourier neural operators from a spectral perspective

---

> > ### Author Rebuttal · Reviewer_6eCN · 2026-04-02
> >
> > I thank the authors for correcting the theoretical part and for providing an additional ablation.
> >
> > I, myself, cannot identify any additional issues with the theory, besides the questionable "naturalness" assumption (which is subjective). I will increase my score accordingly.

---

> > > ### Author Response · Authors · 2026-04-02
> > >
> > > We sincerely thank the reviewer for acknowledging our rebuttal and for kindly raising their score. We truly appreciate their detailed and constructive feedback that has made our theoretical analysis more rigorous.

---

### Official Review · Reviewer_cwx2 · 2026-03-13

**Soundness:** 3
**Presentation:** 2
**Significance:** 2
**Originality:** 2
**Overall Recommendation:** 4
**Confidence:** 3

**Summary:**

This paper proposes IRNO, an iterative refinement framework that improves a pre-trained neural operator through repeated residual correction at test time. The main goal is to mitigate the spectral bias of single-pass neural operators, particularly their difficulty in reconstructing high-frequency details. To do this, the method combines multi-step supervision, progressive spectral loss, and fixed-point regularization, and the experiments on TR-2D, Active Matter, and ERA5 show consistent improvements over several base operators, especially in the mid-to-high frequency regime.

**Compliance With Llm Reviewing Policy:**

Affirmed.

**Final Justification:**

The rebuttal addressed several of my main concerns, and although broader validation would further strengthen the paper, I now revise my recommendation to weak accept.

**Key Questions For Authors:**

1. Have the authors tested refinement backbones that are more operator-native or mesh-aware than U-Net, particularly for irregular-domain settings?

2. Does the method also work well on irregular meshes or more geometry-aware operator settings?

3. Can the method use an adaptive stopping rule instead of a fixed number of refinement steps?

**Limitations:**

See the weaknesses and questions above.

**Strengths And Weaknesses:**

**Strengths**
1. The method is simple and modular, since it can improve a frozen base operator without retraining the whole model.

2. The frequency-domain analysis strongly supports the main claim that the method improves high-frequency errors.

3. The ablation studies on spectral loss and step size are meaningful and help justify the design choices.

**Weaknesses**
1. The theory relies on fairly strong local assumptions, so its relevance to practical finite-width training remains somewhat unclear.

2. The experimental scope is somewhat limited. Although the chosen benchmarks are reasonable, they are still mostly standard structured-grid settings. The paper would be substantially stronger with evidence on irregular meshes, more complex geometries, inverse problems, or larger-scale 3D systems.

3. Similarly, while the framework is described as architecture-agnostic, the actual refinement module used in experiments is U-Net-based, so the demonstrated version remains closely tied to structured-grid settings.

4. The gains become smaller when the base model is already strong, suggesting the method may work best as a correction layer for weaker predictors.

---

> ### Author Rebuttal · Authors · 2026-03-31
>
> We thank reviewer cwx2 for the constructive feedback. Full supporting material is available at [link](https://anonymous.4open.science/r/ICML-IRNO-Rebuttal-84C3/README.md).
>
> ---
> **W1. Theory locality and unclear practical implications**
>
> We agree that the analysis is local and will clarify this is an intentional design. IRNO refines a pre-trained base operator that provides a reasonable ansatz $h_0=T_{\mathrm{base}}(x)$. Assumption 3 requires this initialization to lie within the basin of attraction, which is consistent with a refinement module trained to make stable corrections rather than solve the system from scratch.
>
> To demonstrate the practical breadth of this basin, we conduct a robustness experiment in which the initialization produced by base operator is corrupted by SNR-normalized Gaussian noise with noise levels $\varepsilon$ from 1 to 2. The success ratio of $74.2$% at $\varepsilon=2.0$ (where noise magnitude is twice the signal) suggests that the basin of attraction is a robust feature of the learned landscape.
> ||$\varepsilon=1.00$|$\varepsilon=1.25$|$\varepsilon=1.50$|$\varepsilon=1.75$|$\varepsilon=2.00$|
> |--|--:|--:|--:|--:|--:|
> |success ratio|$100.0$%|$99.9$%|$96.9$%|$86.7$%|$74.2$%|
>
> For finite-width training, the analysis governs once the contractive map is learned. Our empirical Jacobian measurements show that finite-width training can realize this regime. We measure $m$, the minimum eigenvalue of the symmetric part of the Jacobian, on a 1D synthetic example, TR-2D, and Active Matter. Since $m>0$ indicates local strong monotonicity, these results provide direct empirical support for the stable error contraction predicted by the theory.
>
> |Metric|1D-Synthetic|TR-2D U-Net|Active Matter U-Net|
> |--|--:|--:|--:|
> |$m$ (mean $\pm$ std)|$0.661\pm 0.060$|$13.71\pm 2.41$|$3.01\pm 3.12$ |
> |$m>0$|$100$%|$100$%|$99.5$%|
>
> ---
> **W2&3 and Q1&2. Irregular meshes and mesh-aware refinement module**
>
> We train IRNO with RIGNO-12 [1] as both the base operator and the refinement module on the CE-GAUSS dataset with irregular unstructured meshes. This experiment directly shows IRNO extends to irregular meshes and is not tied to specific architectures.
>
> We apply $K=4$ refinements steps in an autoregressive rollout over 7 time steps with $\alpha = 0.3$.
>
> |Model|$t=1$|$t=2$|$t=3$|$t=4$|$t=5$|$t=6$|$t=7$|
> |--|--:|--:|--:|--:|--:|--:|--:|
> |Base $L^2$ Error (%)|$3.351$|$5.017$|$7.808$|$10.923$|$12.732$|$14.433$|$16.617$|
> |IRNO $L^2$ Error (%)|$2.931$|$4.359$|$6.605$|$8.990$|$10.371$|$11.450$|$13.080$|
> |Improvement|$12.5$%|$13.1$%|$15.4$%|$17.7$%|$18.5$%|$20.7$%|$21.3$%|
>
> IRNO achieves consistent improvement across all rollout steps, with gains compounding over time ($12.5$% $\to$ $21.3$%).
>
> We additionally validate architectural-agnosticism on structured-grid settings by evaluating ResNet, ConvNext, and FNO as drop-in refinement architectures on Active Matter (TFNO base, $K=4$):
>
> |Refinement Architecture|Initial|IRNO|Reduction|
> |--|--:|--:|--:|
> |ResNet |$0.1946$|$0.0430$|$77.46$%|
> |ConvNext|$0.1946$|$0.0438$|$77.43$%|
> |FNO|$0.1946$|$0.0559$|$71.26$%|
>
> All refinement architectures achieve substantial error reduction, suggesting that the iterative refinement mechanism by IRNO is robust across architectural choices.
>
> **W4. Smaller gains when base model is strong**
> We agree that the gain is smaller when the base model is already strong. In particular, the improvement over WDSR is more modest because its baseline performance already leaves less room for improvement. Larger gains appear on relatively weaker predictors, though the base operators remain competitive baselines on their respective tasks. IRNO also offers a built-in trade-off between inference time and precision that single-pass models do not.
>
> ---
> **Q3.Can the method use an adaptive stopping rule instead of a fixed number of refinement steps?**
>
> Yes. An adaptive stopping rule is both feasible and practically useful. The simplest approach is to select the stopping step on a validation set when data permits. In case a validation set is limited or unavailable, we can use the magnitude of the refinement update, $m_k = \lVert\Phi_\theta(x,h_k)\rVert$, as a proxy for convergence. We calibrate a threshold $r_{\mathrm{conv}}$ (median ratio of bias-to-update magnitude) on the training set and stop at step $k$ when $\frac{m_k}{m_0} \le r_{\mathrm{conv}}$.
>
> On Active Matter (TFNO), this adaptive stopping rule selected an average of $14.8$ steps, closely matching the VRMSE optimum at $k=15$. This adds negligible computational overhead as $m_k$ is a byproduct during inference. We will include this as a future practical extension.
>
> ---
> We hope our response has addressed your concerns. Should any questions remain, we are happy to discuss further, and we hope you will consider revising your assessment.

---

> > ### Author Rebuttal · Reviewer_cwx2 · 2026-04-04
> >
> > Thank you for the detailed rebuttal and for addressing several of my concerns, especially through the additional results on irregular meshes and architectural flexibility. One remaining suggestion is that the paper would be stronger with validation on a broader range of benchmark domains to more clearly demonstrate generalization. Overall, I appreciate the authors’ response and will revise my score to Weak Accept.

---

> > > ### Author Response · Authors · 2026-04-04
> > >
> > > We sincerely thank the reviewer for acknowledging our rebuttal and for kindly raising their score. We truly appreciate their detailed and constructive feedback.

---

### Decision · Program_Chairs · 2026-04-30

**Decision:**

Accept (spotlight)

**Comment:**

This paper proposes IRNO, an iterative refinement framework for neural operators that mitigates spectral bias by applying a learned correction module over multiple refinement steps. Reviewers found the paper technically solid and well motivated, with strengths including its simple and modular design, strong empirical improvements across several benchmarks, and solid frequency-domain analysis. The work was also viewed as practically appealing because it can enhance a pre-trained base operator without retraining the full model.

The main concerns centered on the strength and scope of the theoretical assumptions, and the extent to which the gains should be attributed to iterative refinement versus the spectral loss. In my judgment, these concerns were substantially addressed in the rebuttal through clarified assumptions, additional theoretical conditions, new results on irregular meshes, and longer-horizon behavior. The reviewers who raised these issues indicated that their concerns were resolved.

Overall, based on the reviews, discussion, and rebuttal, I recommend acceptance.